# Temporal dynamics of Sertoli and germ cell development in human foetal and prepubertal testis

Iris Sanou[1,2,3,*], Mathangi Lakshmipathi[1,2,*], Lisette K. Schönhage[1,2], Saskia K. M. van Daalen[1,2], Cindy M. de Winter-Korver[1,2], Andreas Meißner[1,2,4], Geert Hamer[1,2], Dirk G. de Rooij[5], Rod T. Mitchell[3,6,‡] and Callista L. Mulder[1,2,‡]

## ABSTRACT

Correct development of the testis serves as a springboard for male fertility in adult life, yet our understanding of the timing of human Sertoli and germ cell maturation and their dynamics is incomplete. To map the developmental timeline of germ cells and Sertoli cells, we analysed an extensive set of human foetal and prepubertal testicular samples [n=48, spanning from 7 post conception weeks (PCW) to 13.5 years of age]. Octamer binding transcription factor (OCT)3/4+ gonocytes were identified as the main source of proliferative germ cells during foetal development, while melanoma associated antigen (MAGE)-A4+ (pre)spermatogonia divided at a slow rate both in utero and during childhood. In samples aged between 4 and 10 years, anti-Müllerian hormone (AMH) expression is reduced and androgen receptor (AR) expression is increased, consistent with maturation of testicular Sertoli cells. Sertoli cell proliferation peaked at 2-2.5 years and gradually declined through early childhood, becoming minimal from the age of 6, coinciding with lumen formation. These data suggest that Sertoli cell maturation precedes the initiation of spermatogenesis well before the start of puberty. Ultimately, this human testicular developmental timeline serves as a reference for the development of in vitro gametogenesis models and paves the way for fertility preservation and restoration for those at risk of infertility.

KEY WORDS: Human testis development, Male infertility, Lumen formation, Sertoli cell, Spermatogenesis initiation

[1]Reproductive Biology Laboratory, Centre for Reproductive Medicine, Amsterdam UMC location University of Amsterdam, 1105 AZ Amsterdam, The Netherlands. [2]Amsterdam Reproduction and Development research institute, 1105 AZ Amsterdam, The Netherlands. [3]Centre for Reproductive Health, Institute for Regeneration and Repair, The University of Edinburgh, Edinburgh EH16 4UU, UK. [4]Department of Urology, Centre for Reproductive Medicine, Amsterdam UMC location University of Amsterdam, 1105 AZ Amsterdam, The Netherlands. [5]Reproductive Biology Group, Division of Developmental Biology, Department of Biology, Faculty of Science, Utrecht University, 3584 CH Utrecht, The Netherlands. [6]Department of Paediatric Endocrinology, Royal Hospital for Children and Young People, Edinburgh EH16 4TJ, UK.
*These authors contributed equally to this work

‡Authors for correspondence (c.l.mulder@amsterdamumc.nl; rod.mitchell@ed.ac.uk)

I.S., 0000-0002-7651-8514; M.L., 0000-0002-5081-9597; L.K.S., 0009-0005-1881-3370; A.M., 0000-0002-6200-7553; G.H., 0000-0002-9583-6796; D.G.d.R , 0000-0003-3932-4419; R.T.M., 0000-0003-2501-3003; C.L.M., 0000-0001-9160-7585

## INTRODUCTION

Male reproductive capacity in humans is not solely established by post-pubertal events but is shaped during foetal life after the bipotential gonad commits to a testicular fate. Developmental processes shaping the testis are tightly regulated by signalling cascades, cellular migration and the coordinated maturation and function of germ cells and somatic cells, in which Sertoli cells play a central role. The interplay between these cell types begins during foetal development and continues into puberty, driving key processes such as testis morphogenesis, germ cell differentiation, and the initiation of spermatogenesis (Neto et al., 2021; Walker, 2021; Skinner et al., 1985; Franca et al., 2016). Yet, important gaps in knowledge remain regarding the timing of these early events, which limit progress in developing strategies for fertility preservation and restoration, in children requiring gonadotoxic treatments (Duffin et al., 2024a).

From previous studies using scarce human foetal tissue, it is known that testicular differentiation begins around post conception week (PCW) 6 with the onset of sex-determining region Y (SRY) expression, which triggers the activation of SRY-box transcription factor 9 (SOX9) by PCW 7, marking the formation of premature Sertoli cells (Hanley et al., 2000; Mamsen et al., 2017; Svingen and Koopman, 2013). At this stage, foetal Sertoli cells surround the primordial germ cells (PGCs), which have migrated into the genital ridge, to initiate the formation of testicular cords (Brennan and Capel, 2004; Wilhelm et al., 2007), the precursors of seminiferous tubules. Once enclosed within the cords, PGCs are termed gonocytes, characterised by a high proliferative activity, and subsequently to pre-spermatogonia, marked by the gain of additional germline-specific markers (Culty, 2013; Rajpert-De Meyts et al., 2004; Gaskell et al., 2004; Anderson et al., 2007; Mitchell et al., 2008).

In mice, a similar progression occurs in which migrating PGCs become enclosed within testicular cords, where they are termed gonocytes, maintaining proliferative capacity until around embryonic day 15, before entering quiescence (Kluin and de Rooij, 1981). In humans and non-human primates, it has been reported that germ cells and Sertoli cells exhibit high mitotic activity during foetal and early postnatal life (Schmahl et al., 2004; Schmahl et al., 2000; Franke et al., 2004; Mitchell et al., 2008; McKinnell et al., 2013). During this period, the germ cells remain as a heterogeneous population, as opposed to the synchronous development of germ cells from gonocytes to spermatogonia in rodents. This represents an important difference in testicular development between mice and men. The proliferative activity of Sertoli cells and germ cells during foetal and early postnatal life is likely to have a critical role in future fertility in males and warrants more in-depth investigation across human testicular development.

During human postnatal development, Sertoli cells lose their proliferative potential at an as-yet undetermined time point (Sharpe et al., 2003). In rodents, distinct phases of Sertoli cell proliferation

are well defined (Vergouwen et al., 1991), whereas in humans, it remains unclear whether a second proliferative wave occurs during peri-puberty. Evidence from pre- and peri-pubertal human testicular xenografts showing SOX9/Ki67 co-expression suggests that proliferative activity may persist in Sertoli cells during this developmental window (Hutka et al., 2020). The proliferative arrest is a critical event, as final Sertoli cell number correlates directly with daily sperm output in adulthood (Johnson et al., 1984; Johnson et al., 2008; Sharpe et al., 2003). There are also supportive data showing that the maturation of Sertoli cells is associated with their secretion of anti-Müllerian hormone (AMH). During foetal and early postnatal life, Sertoli cells secrete AMH, which actively causes the regression of the Müllerian ducts and drives testicular development (Josso et al., 1993; Bergada et al., 2006; Aksglaede et al., 2010; O'Donnell et al., 2022). These cells undergo further maturation, characterised by loss of AMH and rise in expression of androgen receptor (AR), which in humans appear to occur largely independently of testosterone (Hero et al., 2012; Rajpert-De Meyts et al., 1999). However, the precise timing of downregulation of AMH or upregulation of AR in the human testis also remains poorly characterised. Expression of AR is crucial, as without it meiosis cannot proceed, which affects spermatogenic potential in post-pubertal life, despite the presence of hormonal cues from the hypothalamus-pituitary-gonadal (HPG) axis (Larose et al., 2020; Chemes et al., 2008).

Despite evidence from animal models and limited human studies suggesting a complementary relationship between Sertoli and germ cell development during early testicular development (Griswold, 1998; Liu et al., 2024; Guo et al., 2020; Willems et al., 2010; Ohta et al., 2012; Yan et al., 2020), knowledge of the temporal dynamics of their maturation in humans remain fragmented, largely due to limited tissue availability and lack of multiplex analytical approaches.

To address these challenges, we developed a custom multiplex immunofluorescence approach to explore the maturation and proliferation dynamics of Sertoli and germ cells from early foetal life through prepubertal life up to 13.5 years of age. This analysis, based on the most extensive human testicular sample collection reported to date, provides crucial insights into early testicular development, and lays the foundation for improving fertility preservation strategies for prepubertal male patients at risk of losing their fertility.

## RESULTS

### Morphogenesis and development of testicular cords in the human foetal and prepubertal testis

We analysed a cohort of 48 human testicular samples (34 foetal and 14 postnatal) to assess morphogenesis during testis development. In the earliest sample of PCW 7, the testicular architecture appeared as disorganised clusters of germ cells, Sertoli cells and other somatic cells, lacking a distinct cord-like morphology. From PCW 8-9 onwards, testicular cords were observed to be composed of a mixture of Sertoli and germ cells (Fig. 1A, PCW 7, PCW 8). These cords were encapsulated by elongated peritubular myoid cells providing structural support.

From the formation of cords from PCW 8-9, the testes appeared distinctly compartmentalised into cords and interstitial regions. This was accompanied by a marked increase in cord area, which was maintained throughout development with minimal variations in the relative proportions of cord and interstitial areas across foetal samples (Fig. 1B). Concurrently, cord diameter increased steadily until PCW 12, after which it plateaued at approximately 50 μm through to 8 years of age (Fig. 1C,E; Table S1). In contrast,

prepubertal testes displayed greater variability in the relative proportions of seminiferous compartment and interstitial areas, although this may, in part, be attributable to the low number of biological replicates available for these samples (Fig. 1D). This suggests that longitudinal growth of the seminiferous compartment may continue independently of radial growth during this phase (Malas et al., 2001; Nistal et al., 1982). Importantly, lumen formation within the seminiferous compartment was first evident around 6.5 years of age (Fig. 1A, 6.5 years), becoming more prominent from 8.5 years onwards. From around 8.5 years onwards, tubular diameter increased markedly from 65 μm, reaching 120 μm by 13.5 years and showed a strong positive correlation with age (Fig. 1E). The range of developing germ cell types observed increased with age, with spermatocytes and spermatids appearing at around 10 and 13.5 years of age, respectively (Table S2). This observed postnatal increase in tubular diameter reflects the onset of pubertal testicular growth (Amer et al., 2022), consistent with progressive expansion of tubular structures and increased complexity of germ cell populations observed in our samples. Collectively, these findings demonstrate structural development of the seminiferous cords during foetal and prepubescent life, thereby establishing the specialised microenvironment for onset of spermatogenesis upon puberty.

### Temporal dynamics of AMH and AR expression in Sertoli cells

SOX9 was detectable in the Sertoli cells as early as PCW 7, preceding the formation of testicular cords, and remained consistently expressed up to PCW 21, as well as postnatally (Fig. 2A; Fig. S1A, Tables S3 and S4). At PCW 7, SOX9+ cells appeared disorganised due to the absence of cord structures. However, by PCW 8-9, cord formation had been initiated and Sertoli cells displayed a more defined organisation within seminiferous cords (Fig. 2A, PCW 8). From approximately PCW 14 onward, a clear reorganisation of the seminiferous cords became apparent as the Sertoli cell nuclei, marked by SOX9 expression, were increasingly aligned along the basal membrane (Fig. 2A, PCW 21, magenta arrowheads; Fig. S1B). This spatial reorganisation indicates the onset of cellular polarity within the cords, a key feature of progressing testicular development as previously described in rats (Magre and Jost, 1991). This finding provides the first developmental time point for the onset of Sertoli cell polarisation in the human foetal testis.

Similar to SOX9, AMH expression was also detected from the earliest sample at PCW 7 with its spatial distribution completely overlapping SOX9+ Sertoli cells (Fig. 2A; Fig. S1A), consistent with prior findings of AMH induction around PCW 7-8 (Josso et al., 1993; Ostrer et al., 2007). Early AMH expression is stimulated by transcriptional regulation through SOX9, together with other key factors such as WT1 and NR5A1 (De Santa Barbara et al., 1998; Nachtigal et al., 1998). Functionally, AMH is crucial for Müllerian duct regression and serves as a marker of functional immature Sertoli cells. From PCW 14 onwards, AMH, like SOX9, became progressively restricted to the basal region, suggesting the onset of Sertoli cell polarisation during this developmental window. This basal restriction became progressively more pronounced with advancing foetal age (Fig. 2A, PCW 21, yellow arrowheads; Fig. S1B), with AMH expression remaining consistently high up to the oldest examined foetal age of PCW 21 (Fig. 2A,C; Tables S3 and S4).

Expression of AR, a marker for Sertoli cell maturation, but also expressed in Leydig cells and peritubular myoid cells, was detected at PCW 7-8 only in the interstitial compartment, localising to Leydig and peritubular myoid cells. AR expression in these cell

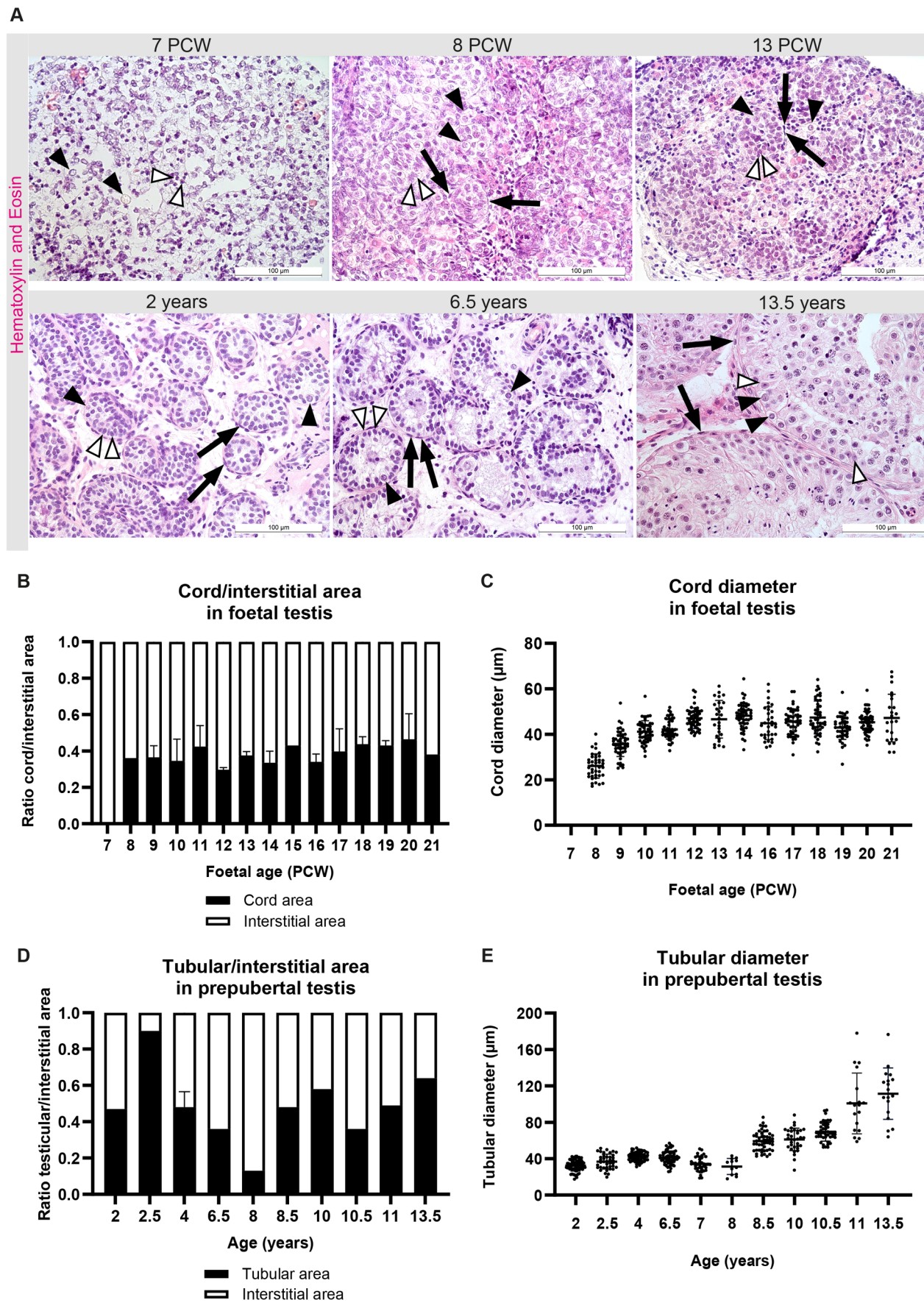

**Fig. 1.** See next page for legend.

**Fig. 1. Haematoxylin and Eosin (H&E) staining of testicular tissue sections across developmental stages.** Seminiferous cord/tubule formation in foetal and prepubertal testicular sections. (A) Representative overview images of H&E-stained testicular tissue sections of foetal testes at PCW 7, 8, 13, and prepubertal testes aged 2, 6.5 and 13.5 years. Closed arrowheads indicate germ cells, open arrowheads indicate Sertoli cells, and arrows indicate peritubular myoid cells. Scale bars: 100 µm. (B) Quantification of cord/interstitial area in the developing human foetal testis. (C) Cord diameter in the developing human foetal testis; each dot represents the diameter of a seminiferous cord. (D) Quantification of tubular/interstitial area ratio in the developing human prepubertal testis. (E) Tubule diameter in the developing human prepubertal testis; each dot represents the diameter of a seminiferous tubule. Data shown are mean±s.d. (*n*=3 for PCW 11, 12 and 16-20; *n*=2 for PCW 7, 9, 10, 13 and 14, 4 years, 6.5 years and 13.5 years; *n*=1 for PCW 8, 15 and 21, 2 years, 2.5 years, 7 years, 8 years, 8.5 years, 10 years, 10.5 years and 11 years).

types was maintained throughout development and persisted into adulthood, up to the oldest age examined (13.5 years). In contrast, AR expression within the testicular cords was minimal throughout foetal life, with sporadic faint expression of AR colocalising with SOX9 expression (Fig. 2B,C; Table S3), suggesting an immature Sertoli cell phenotype in the majority of these cells, consistent with robust AMH expression during the foetal period.

Postnatally, AMH was widely expressed in Sertoli cells between 2 and 8.5 years of age, followed by a sharp decline after 8.5 years (Fig. 2A,C; Fig. S3). This downregulation during the peri-pubertal phase is consistent with previous findings showing a concurrent increase in intratesticular testosterone levels around this age (Lee et al., 1996), although the exact timing and magnitude of this rise varies among individuals. Notably, during early postnatal life, AMH expression in Sertoli cells is stimulated by follicle-stimulating hormone (FSH), which promotes Sertoli cell proliferation and sustained AMH secretion in childhood (Edelsztein et al., 2022). In contrast, AR was not detected in Sertoli cells during the early postnatal period but began to increase gradually from around 4 years of age, suggesting the initiation of phenotypic maturation of Sertoli cells (Fig. S1C and Fig. S3). This acquisition of AR expression may also reflect increasing Leydig cell activity and androgen production during prepubertal development. We observed that AR expression in Sertoli cells exhibited a sustained increase correlated to age, whilst the decrease observed in the sample from an 8-year-old likely reflects inter-sample variability. This gradual increase in AR expression aligns with progressive Sertoli cell maturation occurring during early puberty. Notably, this sustained AR expression coincided with a sharp decline in AMH from 8.5 years of age, after which AMH remained undetectable (Fig. 2D; Fig. S3, top row, Tables S3 and S4). This reciprocal pattern of declining AMH expression with increasing AR expression strongly supports the transition toward Sertoli cell maturation and the onset of pubertal testicular function (Boukari et al., 2009; Lapoirie et al., 2021). It should be noted that these timings are approximate, based on a limited number of samples, and considerable individual variation in maturation and onset of puberty is expected (Brix et al., 2019).

### Gonocyte maturation to pre-spermatogonia in the human foetal testis

In the earliest examined sample (PCW 7), OCT3/4 expressing gonocytes were dispersed throughout the testis, in the absence of defined tubule structures (Fig. 3A, PCW 7). From PCW 8-9 onwards, OCT3/4 expression continued to be restricted to germ cells located within the organising seminiferous cords. These OCT3/4+ gonocytes were distributed throughout the developing cords, indicating a lack of spatial orientation early in development (Fig. 3A, PCW 9).

Notably, MAGE-A4+ pre-spermatogonia were first observed sporadically within the testicular cords until PCW 10 (Fig. 3A,C; Tables S4 and S5). Compared to OCT3/4+ gonocytes, these cells tend to be larger, with rounder nuclei and typically localised at the basal side (Fig. S2A). This presence of MAGE-A4 is largely linked to the loss of pluripotency marker OCT3/4, consistent with previous observations (Guo et al., 2021). From PCW 10-11 onwards, MAGE-A4+ pre-spermatogonia gradually became more frequent, coinciding with a continued presence of OCT3/4+ gonocytes (Fig. 3A,C), thereby suggesting a progressive shift in germ cell maturation. Interestingly, a small population of germ cells co-expressing OCT3/4 and MAGE-A4 was also occasionally observed from PCW 9 up to our latest gestational age examined (PCW 21) (Fig. 3B, white arrowheads; Table S5), consistent with prior reports (Mitchell et al., 2008, 2010). This likely represents a brief transitional state during germ cell differentiation, also considering the short half-life of the OCT3/4 protein (Gaskell et al., 2004; Pauls et al., 2006; Saxe et al., 2009). Contrary to OCT3/4+ gonocytes, which were typically located in the centre of the cords with their nuclei positioned away from the basal membrane, MAGE-A4+ pre-spermatogonia were typically found localised at the basal membrane, maintaining their nuclei in close apposition to it (Fig. 3A; Fig. S2B). Occasionally, cells within the cords that were devoid of OCT3/4 or MAGE-A4 were observed, potentially representing a transitionary germ cell state (Overeem et al., 2021).

After birth, from the youngest postnatal sample of 2 years that was included in this study, OCT3/4+ gonocytes were no longer detected within the seminiferous tubules. This was consistent with the previously reported complete downregulation of OCT3/4 and maturation towards a committed spermatogonial fate (Rajpert-De Meyts et al., 2016). Concurrently, the number of MAGE-A4+ spermatogonia per tubular area increased substantially with increasing postnatal age (Fig. 3D; Tables S4 and S5), where their nuclei were predominantly localised at the basement membrane (Fig. 3A, 2.5 and 4 years; Fig. S3, bottom row). Occasionally, MAGE-A4+ spermatogonia were also detected in the ad luminal regions of the tubules in samples up to 4 years of age (Fig. 3A; Fig. S2B). However, at 10.5 years in our sample cohort, MAGE-A4+ spermatogonia were consistently and exclusively located adjacent to the basement membrane of the tubules (Fig. 3A, 13.5 years; Fig. S3, bottom row).

### Germ-Sertoli cell proliferation patterns in the human developing testis

The percentage of proliferative cells within the OCT3/4+ cell population, ranged between 4% and 41% across samples, varying considerably between samples (Fig. 4A,D; Table S4), in line with previous reports (Gaskell et al., 2004; Pauls et al., 2006; Saxe et al., 2009). At the earliest stages (PCW 7–9), while MAGE-A4+ cells started to emerge, no Ki67/MAGE-A4 co-expressing cells were detected. Proliferative MAGE-A4+ pre-spermatogonia first appeared during foetal development at PCW 10, while the proportion of MAGE-A4+ proliferative cells was substantially lower than that of OCT3/4+ proliferative cells (Fig. 4B,D). Postnatally, proliferative MAGE-A4+ spermatogonia were detected sporadically between 2 and 6.5 years. From 8 years onward, a rise (24%) was detected (Fig. 4D). No cells co-expressing OCT3/4, MAGE-A4, and Ki67 were identified in this cohort.

Regarding Sertoli cell proliferation, we observed low proliferative activity in foetal life (0-11%, PCW 7-21), while in our early postnatal samples, proliferation increased to 24-36% of the entire Sertoli cell population (Fig. 4C), linking to expansion and organisation of seminiferous cords and the increase in tubule diameter (Fig. 1D). In

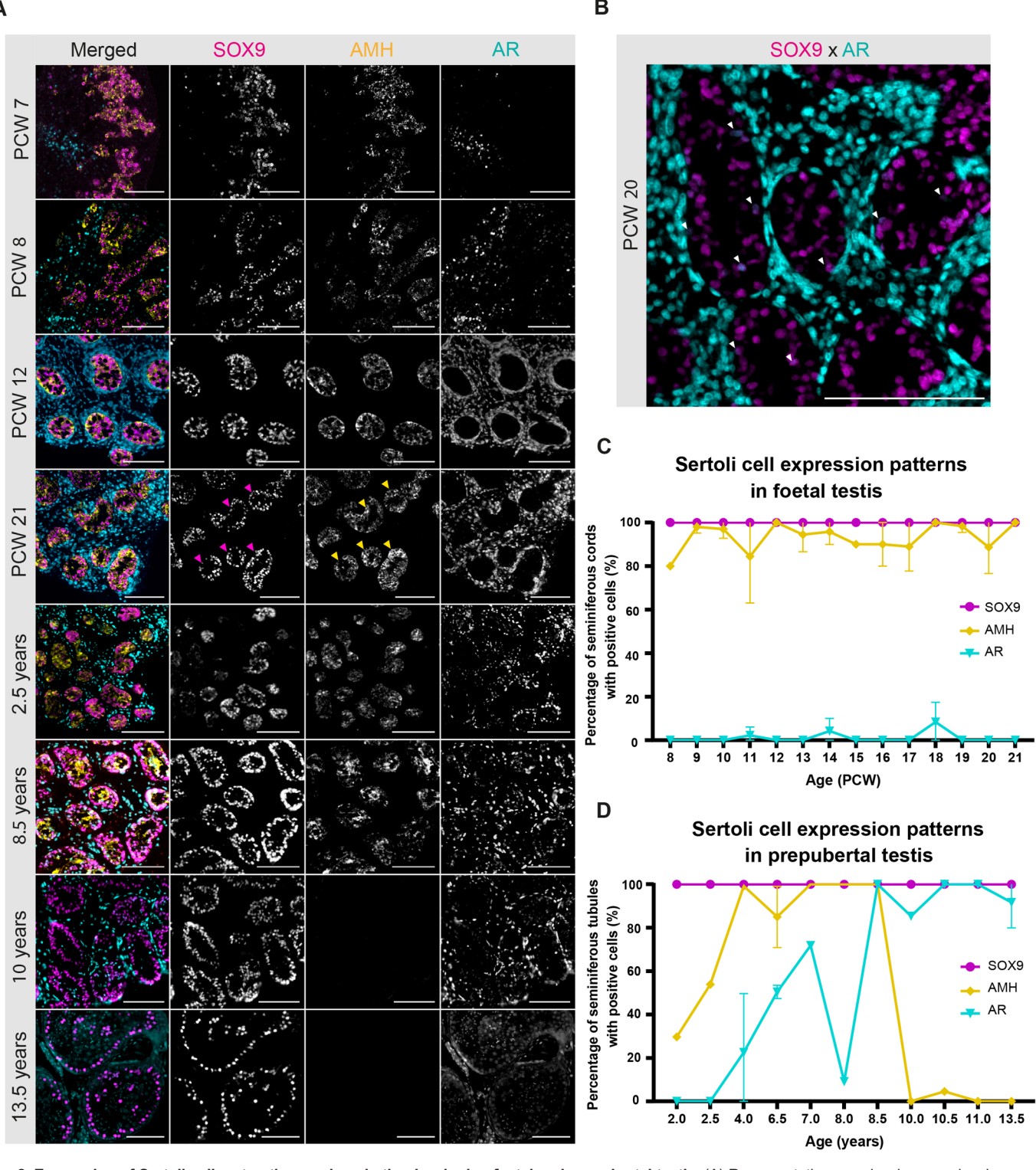

**Fig. 2. Expression of Sertoli cell maturation markers in the developing foetal and prepubertal testis.** (A) Representative overview images showing expression of SOX9 (magenta), AMH (yellow) and AR (cyan) in foetal testes at PCW 7, 8, 12 and 21, and in prepubertal testes aged 2.5, 8.5, 10 and 13.5 years. Magenta and yellow arrowheads indicate the polarisation of Sertoli cells, marked by SOX9 and AMH, respectively, during this developmental window. Scale bars: 100 µm. The (merged) images of PCW 7 and PCW 21 are also shown in Fig. S1A,B, respectively, and (merged) images of 2.5 years and 10 years are also shown in Fig. S3. (B) Higher-magnification image from a PCW 20 foetal testis showing SOX9 (in magenta) and AR (in cyan) co-expression, with white arrowheads indicating SOX9 and AR double-positive Sertoli cells. Scale bar: 100 µm. (C) Percentage of seminiferous cords positive for each marker (SOX9, AMH, and AR) across foetal testis development. (D) Percentage of seminiferous tubules positive for each marker (SOX9, AMH, and AR) across prepubertal testis development. Data shown are mean±s.d. (n=3 for PCW 11, 12 and 16-20; n=2 for PCW 7, 9, 10, 13 and 14, 4 years, 6.5 years and 13.5 years; n=1 for PCW 8, 15 and 21, 2 years, 2.5 years, 7 years, 8 years, 8.5 years, 10 years, 10.5 years and 11 years).

Biology Open

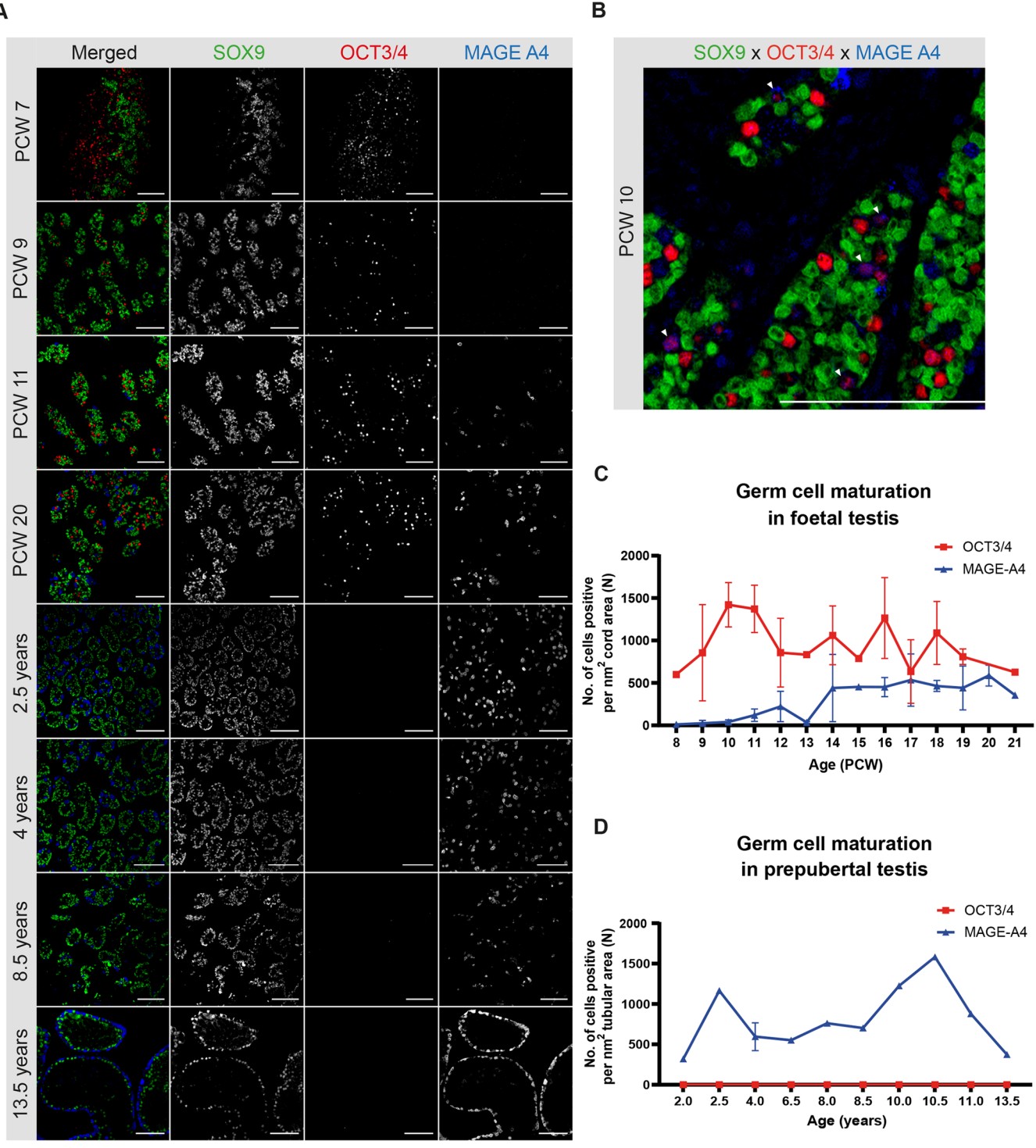

**Fig. 3. Expression of germ cell maturation markers in the developing foetal and prepubertal testis.** (A) Representative images showing SOX9 (in green), OCT3/4 (red) and MAGE-A4 (blue) expression in foetal testes at PCW 7, 9, 11 and 20, and in prepubertal testes aged 2.5, 4, 8.5, and 13.5 years. Scale bars: 100 µm. The (merged) image of 2.5 years is also shown in Fig. S2B, and (merged) images of 2.5 years and 4 years are also shown in Fig. S3. (B) Higher-magnification image from a PCW 10 foetal testis showing SOX9 (green), OCT3/4 (red), and MAGE-A4 (blue), with white arrowheads indicating OCT3/4 and MAGE-A4 double-positive germ cells. Scale bar: 100 µm. (C) Quantification of OCT3/4 and MAGE-A4 single-positive germ cells per cord area (nm²) in the foetal testis. (D) Quantification of OCT3/4 and MAGE-A4 single-positive germ cells per tubular area (nm²) in the prepubertal testis. Data shown are mean±s.d. ($n$=3 for PCW 11, 12 and 16-20; $n$=2 for PCW 7, 9, 10, 13 and 14, 4 years, 6.5 years and 13.5 years; $n$=1 for PCW 8, 15 and 21, 2 years, 2.5 years, 7 years, 8 years, 8.5 years, 10 years, 10.5 years and 11 years).

our sample set, from 4 years onwards, Sertoli cell proliferation was minimal and became negligible after 10.5 years of age. A correlation between the loss of Sertoli cell proliferation, marking a terminal state of maturation, and the shift to AMH loss and concurrent AR gain between ages 4-10 years in our cohort show that these events coincide developmentally. In addition, lumen formation was first observed at

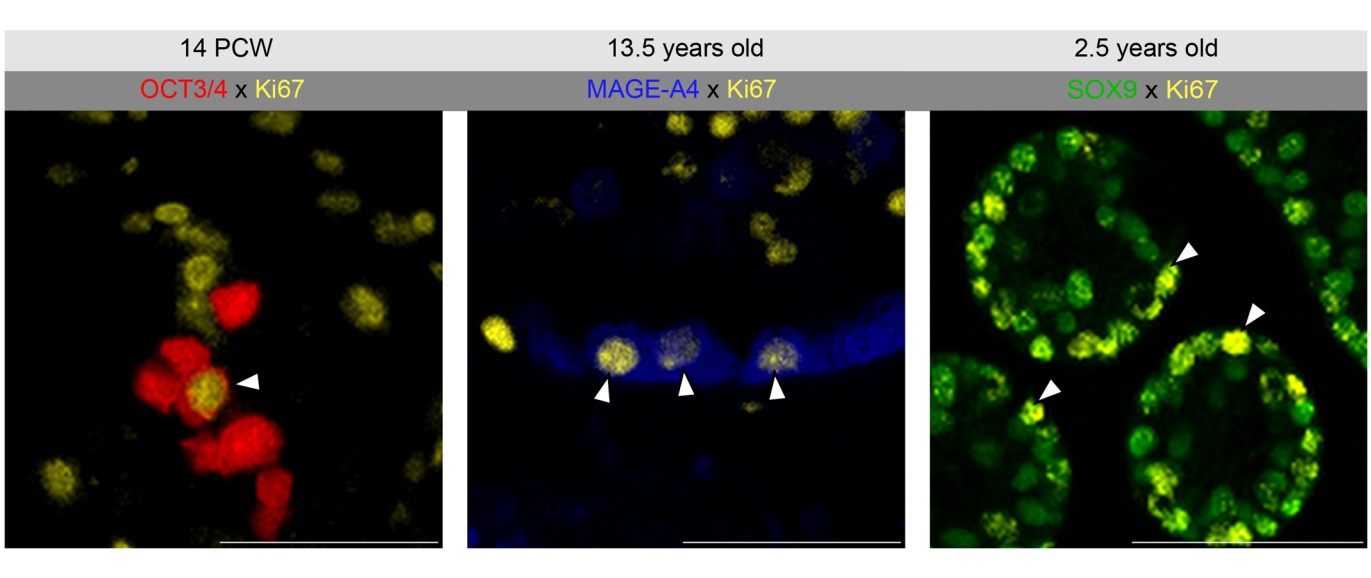

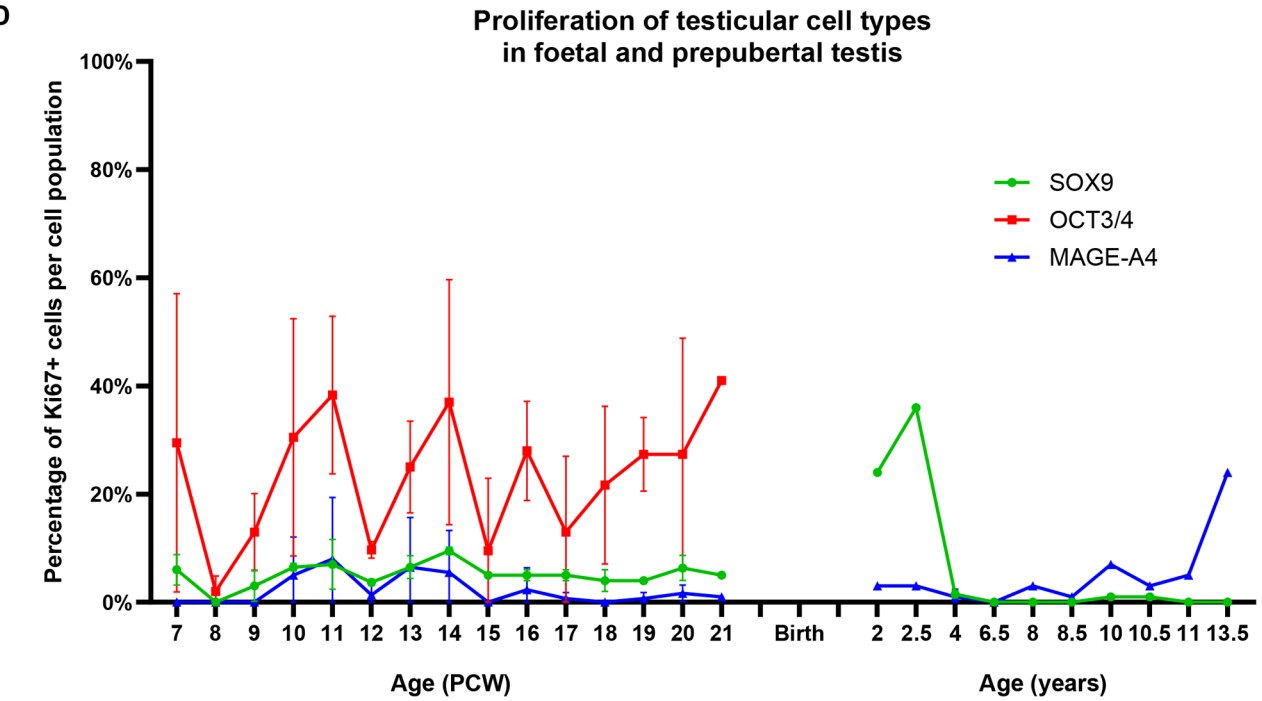

**Fig. 4. Proliferation of germ cells (OCT3/4 and MAGE-A4) and Sertoli cells (SOX9) in the developing human testis.** (A) Representative image showing co-staining of OCT3/4 (red) and Ki67 (yellow) from a PCW 14 foetal testis, with white arrowheads indicating double-positive cells. Scale bar: 50 µm. (B) Representative image showing co-staining of MAGE-A4 (blue) and Ki67 (yellow) from a prepubertal testis aged 8.5 years, with white arrowheads indicating double-positive cells. Scale bar: 50 µm. (C) Representative image showing co-staining of SOX9 (green) and Ki67 (yellow) from a PCW 16 foetal testis, with white arrowheads indicating double-positive cells. Scale bar: 50 µm. (D) Percentage of proliferative Ki67+ cells within the SOX9, OCT3/4 and MAGE-A4 cell populations. Data shown are mean±s.d. (n=3 for PCW 11, 12 and 16-20; n=2 for PCW 7, 9, 10, 13 and 14, 4 years, 6.5 years and 13.5 years; n=1 for PCW 8, 15 and 21, 2 years, 2.5 years, 7 years, 8 years, 8.5 years, 10 years, 10.5 years and 11 years).

6.5 years and became prominent from 8.5 years onwards, potentially priming the testicular tissue for puberty onset. Collectively, these results reveal dynamic patterns of how the foetal and prepubertal testicular somatic environment prepares for germ cell differentiation and spermatogenesis.

**A timeline of maturation dynamics in the human testis**
Based on this specific cohort of samples (n=48), the developmental progression of the human testis is characterised by key landmarks

preparing the somatic niche prior to the initiation of spermatogenesis at puberty. Early foetal life features the formation of testicular cords, harbouring proliferative Sertoli and germ (OCT3/4+ and, later, MAGE-A4+) cells. From 2 years of age, MAGE-A4+ spermatogonia were the only germ cell population present. Sertoli cells remain immature before birth, with only sporadic AR expression seen. From early childhood, Sertoli cells undergo a dynamic process beginning with relocalisation and a peak in proliferation, followed by lumen formation and a complete transition in marker expression from AMH

to AR, marking their terminal differentiation at approximately age 10 in this cohort. These milestones collectively establish the testicular somatic environment that supports the onset of spermatogenesis during puberty (Fig. 5).

## DISCUSSION

While foetal development represents a critical window of testicular morphological compartmentalisation, the prepubertal phase after the infantile period (after approximately 1.5 years of age) has generally been viewed as a period of relative quiescence (Sharpe et al., 2003). We demonstrate that the somatic niche of the human prepubertal testis initiates preparation for spermatogenesis earlier than previously described. This knowledge is relevant for fertility preservation in boys that need to undergo severe gonadotoxic treatments (ESHRE FP for Boys Working Group et al., 2025; Duffin et al., 2024a).

An environment permissive for the initiation of spermatogenesis, thought to occur around the onset of puberty (Sharpe et al., 2003), may be established years earlier in development, based on our data. The expected loss of AMH expression and gain of AR expression occurred asynchronously within the testis from the age of 4, and is completed in our cohort at the age of 10 years, marking somatic testicular maturation. In addition, our data revealed that the formation of lumen coincides with a decrease in the proliferative potential of Sertoli cells in samples of ages 4 years and up, suggesting that Sertoli

cell numbers are largely established in early childhood. Most of the MAGE-A4+ spermatogonia had migrated to the basal membrane at this point, which may be indicative of ongoing maturation process of the somatic microenvironment. Some MAGE-A4+ spermatogonia, were still detected in the ad luminal regions of the tubules in samples up to approximately 7 years of age, possibly indicating residual or delayed migratory activity during the prepubertal quiescent phase (Paniagua and Nistal, 1984). In our cohort, the assumed onset of puberty is marked by an increase in proliferation of MAGE-A4+ germ cells, in our sample at 13.5 years of age.

The proliferation of spermatogonia is essential for male fertility. A low or absent number of spermatogonia is associated with non-obstructive azoospermia, a severe form of male infertility in which no sperm is found in the ejaculate. Proliferation of germ cells has been studied during foetal life in non-human primates (Mitchell et al., 2008), with a lower proliferation index for germ cells compared to postnatally. In addition, a high proliferation index was attributed to OCT3/4+ germ cells in a human-to-mouse xenotransplantation study using samples aged PCW 9 and PCW 14-18 (Mitchell et al., 2010). Separate analysis of proliferative OCT3/4+ gonocytes and MAGE-A4+ pre-spermatogonia over a weekly time-course from PCW 7 to PCW 21 was not done before. Using multiplex staining, we found that a large proportion of OCT3/4+ gonocytes show active proliferation in foetal life, while MAGE-A4 expressing pre-spermatogonia exhibited low

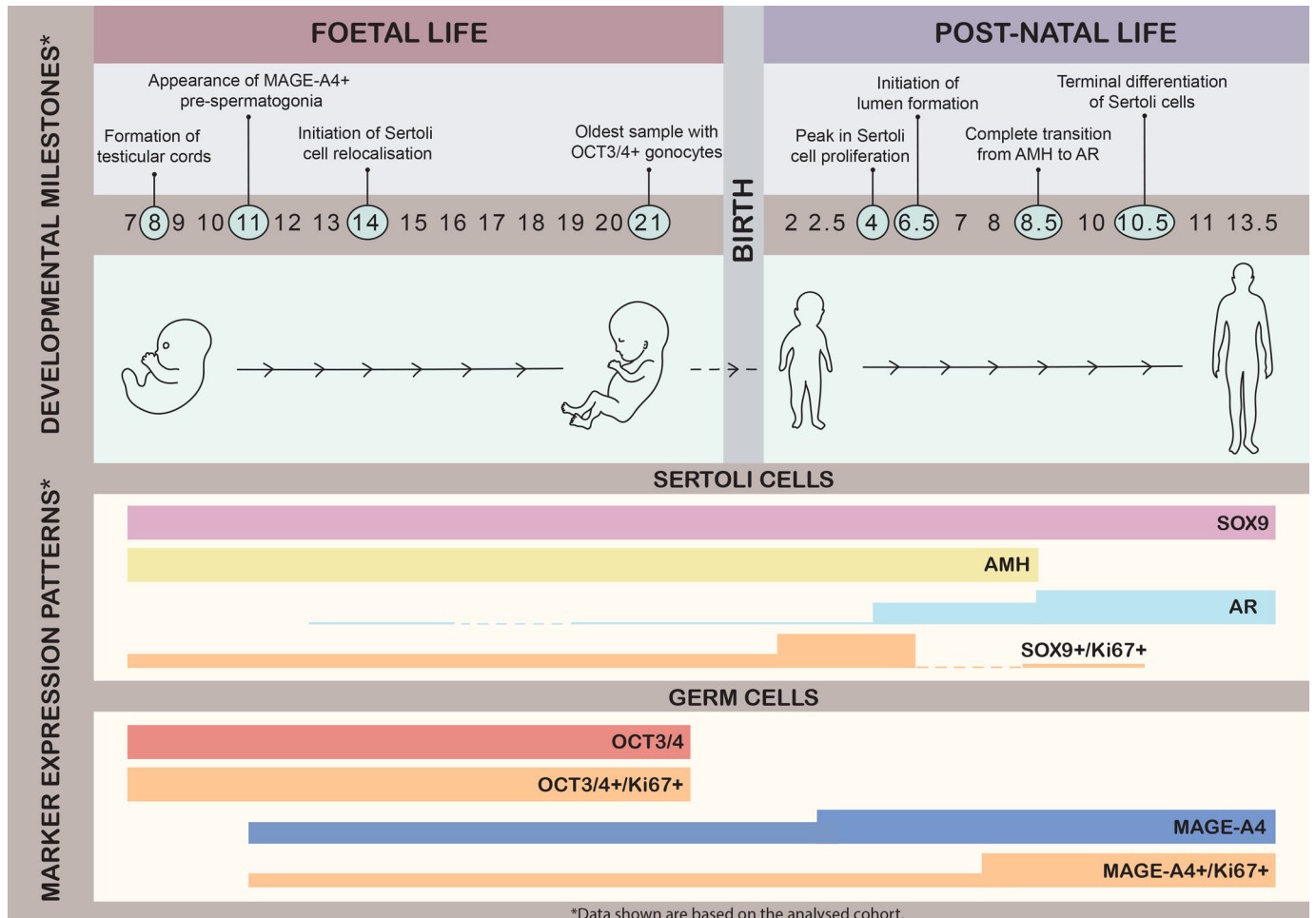

**Fig. 5. Developmental timeline and maturation dynamics in the human testis.** A schematic representation of key developmental milestones from early foetal life through prepuberty, highlighting the establishment and maturation of the Sertoli and germ cells as reflected by their characteristic gene expression patterns.

proliferative activity during both foetal and prepubertal life. This may indicate that the bulk of the germ cell population is established from foetal gonocytes, while limited proliferation of MAGE-A4 spermatogonia further expands the spermatogonial population.

We only observed OCT3/4 expression in foetal tissues in our cohort, supporting the concept that gonocytes are fully differentiated into spermatogonia by the age of 1 year (Rajpert-De Meyts et al., 2004; Honecker et al., 2004; Mitchell et al., 2014). This developmental transition is clinically significant, as a failure to downregulate pluripotency markers such as OCT3/4 and mature into (pre)spermatogonia has been implicated in the pathogenesis of testicular germ cell neoplasia *in situ*, the precursor for testicular cancer (Rajpert-De Meyts et al., 2016). The timing of loss of OCT3/4 expression is, however, asynchronous and reported to occur over an extended developmental window, spanning from the first trimester through to approximately 6 months postnatally (Mitchell et al., 2008; Rajpert-De Meyts et al., 2004). Since our sample collection did not contain any samples below the age of 1.8 years, we were not able to study the period of 'mini-puberty', typically ending at the age of 6 months in boys (Rohayem et al., 2024), which is a limitation of this study.

While our study represents the largest human sample cohort both in sample number and completeness of the developmental timeline, there is inter-individual variation in testicular development between samples. An example is the onset of puberty, which typically begins at any time between the ages of 9-14 years (Brix et al., 2019). The same may hold true for earlier childhood and foetal developmental milestones. Definitive confirmation of the timing of these events would require analysis of a larger number of samples at each age than the one presented in this study. It should be noted that sample availability will continue to be limited for late foetal and perinatal ages due to stringent ethical considerations governing the use of human tissues at these developmental stages.

By performing multiplex immunofluorescent assays, allowing direct analysis of key-events in testis maturation, we provide deeper knowledge that connects to existing literature. While several transitions have been established previously using limited human samples, such as the expression of AMH from PCW 7 onwards and the emergence of AR as a Sertoli cell maturity marker (Josso et al., 1993; Bergada et al., 2006; Chemes et al., 2008; Regadera et al., 2001), we were able to show that AR and AMH expression fluctuated more substantially than expected, well before initiation of puberty. Surprisingly, we observed a low expression of the maturity marker AR in a subpopulation of Sertoli cells throughout foetal life, which has not been reported before. A potential explanation may be that these Sertoli cells could show early signs of maturation in response to germ cells that aberrantly attempt to mature precipitously to pre-leptotene spermatocytes (Devictor et al., 1979). A previous study hypothesised that the low AR expression during early postnatal life may serve a biological role in preventing premature Sertoli cell maturation and initiation of spermatogenesis, until the appropriate developmental window (Chemes et al., 2008). This low AR expression could also be driven by low circulating testosterone and luteinising hormone (LH) levels during this period, which are known regulators of Sertoli cell development. In addition, fluctuations of AR during childhood may be concomitant to observed occasional presence of B spermatogonia and early spermatocytes in other sample collections of testicular biopsies of boys aged 4-7 years (Masliukaite et al., 2016). In our foetal and early childhood samples, spermatocytes were not observed. Therefore, AR expression in Sertoli cells may be temporary at these early ages.

While multiplex immunofluorescent assays allowed us to evaluate multiple markers simultaneously, this was restricted to four colours per section, impeding us from studying other markers relevant to Sertoli and germ cell development, which is a limitation of our study. For example, occasionally intratubular cells devoid from all staining were observed, which most likely correspond to a previously reported MAGE-A4−/OCT3/4− double negative germ cell population at 18 PCW (Overeem et al., 2021). Their location within the cords would indicate that this represents a germ cell population, while co-staining of additional germ cell markers (e.g. DDX4) might have been informative to confirm their identity. In addition, our chosen method did not allow for identification of a broader range of factors important for testicular development. We were limited by the sample fixation using Bouin's solution, containing picric acid, known to cause chemical modifications and cross-linking in nucleic acids. This prevented us from analysing more factors effectively in this tissue.

Our findings serve as a valuable reference work for both fundamental research and translational applications with regards to fertility preservation and restoration. Worldwide, fertility clinics are storing tissue from boys that require gonadotoxic treatment (e.g. cancer patients, patients with benign haematological disease) (Duffin et al., 2024b), with the goal of trying to resolve treatment-induced infertility through spermatogonial stem cell based fertility restoration treatments (Duffin et al., 2024a; Sanou et al., 2022). Proper knowledge on the development of the testis will be key in developing these fertility restoration methods. For example, protein expression of markers of Sertoli and germ cell maturity are often used in *in vitro* systems aiming to achieve sperm formation in immature testicular tissue fragments (Younis et al., 2023) or in pluripotent stem cell derived gametogenesis systems (Liang et al., 2019; Rodriguez Gutierrez et al., 2018; Pryzhkova et al., 2022; Knarston et al., 2020). By highlighting the nuanced expression patterns of (im)maturity markers such as AR and AMH in the human testis, this study emphasises the importance of careful interpretation when assessing testicular maturation, especially in experimental or *in vitro* contexts.

In addition, the developmental timings of both the establishment of the germ cell population and somatic niche maturation may be useful in addressing the impact of treatments in early childhood, or even *in utero*, including the treatment of pregnant patients. Reports are emerging that medication that is considered safe may be detrimental to the developing gonad, including paracetamol, a widely used pain medication (Bauer et al., 2021). Currently, chemotherapy treatment for breast cancer in the second or third trimester of pregnancy is considered to be safe for the unborn child (Poggio et al., 2020), while we show that germ cell proliferation is still high in second trimester. Still, very limited data exist on the fertility of children exposed to chemotherapy *in utero*, based on the timing exposure in relation to gonad development (Sebire et al., 2025). Ultimately, identifying critical windows of development of the testis not only enhances the reliability of emerging research models but also paves the way for novel therapies to address male infertility and preserve or protect fertility for those that need it.

## MATERIALS AND METHODS
### Experimental model and subject details
#### Human testicular tissue
For a list of human tissues, see Table S2. All used human (*Homo Sapiens*) foetal testicular tissue samples from first and second trimester (*n*=34, 7-21 PCW) were obtained from medical and surgical terminations of pregnancy on social indication. Informed consent was obtained from the pregnant mother, and the study was conducted in accordance with the ethical principles of the Declaration of Helsinki. Tissue has been utilised for scientific research purposes and is approved by the South-East Scotland Research Ethics Committee (LREC08/S1101/1), NRES committee North-East Newcastle and North Tyneside 1 (08/H0906/21+5) and NRES Committee London–Fulham (18/10/0822).

Biology Open

Usage of all human prepubertal testicular tissue samples ($n$=14, 2-13.5 years) for scientific research purposes was approved by the National Ethical Committee (CCMO) at the Amsterdam UMC/ location AMC (NL27690.000.09), and informed consent was given from patients' parents before obtaining testicular tissue for fertility preservation strategies.

To reduce risk of bias, all samples were assigned randomised identification codes by two independent researchers, anonymising them throughout the experimental procedures. The codes were maintained confidentially and were not accessible to the investigators during data collection and analysis. Codes became accessible after all data had been collected and finalised.

### Method details

#### Human foetal testicular tissue sampling
To extract the testis, an incision was made in the lower abdomen of the foetus. The testes were located in the abdomen or scrotum, depending on the developmental stage of the foetus. Foetal samples were collected and PCR analysis was performed targeting the male-specific *SRY* gene to ascertain sexual genotype (data not shown). None of the terminations included identified foetal abnormalities. The gestational age of each specimen was determined via ultrasound followed by measurement of foot length.

#### Human prepubertal testicular tissue sampling
Male prepubertal patients underwent surgery to obtain a testicular biopsy under general anaesthesia. The majority of each biopsy was cryopreserved and stored for potential future clinical usages for fertility treatment. A portion of around 15% was used for research purposes.

#### Sample fixation
For a list of chemicals and reagents see Table S6. All human foetal samples were fixed for 1 h with Bouin's fluid (Clin-Tech, CTEH64096/1) and flushed multiple times with 70% ethanol (EtOH; 100% EtOH, Merck, 1.00983.1000). Prepubertal samples were fixed with diluted prepared Bouin's [5% acetic acid (Merck, 1.00063.1000), 9% formaldehyde (Merck, 1040031000) and 1.5% picric acid (Sigma-Aldrich, P6744)] for at least 3 h and flushed multiple times with 70% EtOH. All tissues were embedded in paraffin. Multiple serial sections of 5 µm were made for all foetal and prepubertal samples.

#### General morphology
For histological examinations, all specimens were stained using Haematoxylin and Eosin (H&E) to assess and verify the quality of testicular tissue structures before inclusion in the study. The quality assessment was based on the extent of apoptosis, observed as pyknotic areas, in the tissue and quality of the section in relation to artefacts.

Slides were deparaffinised in xylene (VWR, 28975.325). Subsequently, the slides were hydrated in a series of decreasing concentrations of ethanol starting with 100% EtOH and finishing in deionised water. Tissues were stained with Mayer's Haematoxylin (Roth, 517-28-2). Afterward, the samples were rinsed under running tap water. The tissues were then stained with Eosin (VWR, 1.15935.0100) in 70% EtOH, followed by a rinse with deionised water and 70% EtOH. The samples were dehydrated and embedded with Entellan (VWR, 1.07961.0100).

#### Multiplex tyramine staining procedure of Sertoli cell maturation
Consecutive 5-µm sections were stained for Sertoli cell and germ cell markers, respectively, to allow direct comparison of Sertoli cell and germ cell maturation and proliferation for each sample.

For a list of antibodies, see Table S7. Testicular tissue sections were deparaffinised in xylene I and xylene II, each for 5 min, followed by hydration of the tissue sections in a decreasing concentration of EtOH (100% EtOH I, 100% EtOH II, 96% EtOH and 70% EtOH), each for 2 min at room temperature (RT). Post tissue fixation was performed with 4% formaldehyde (ROTH Histofix, P087.4) for 20 min at RT. Slides were washed three times in Tris [2-amino-2-(hydroxymethyl)-1,3-propanediol (Roche, 10708976001)]-buffered saline with Tween [TBT; 0.05% Tween-20 (Sigma-Aldrich, P1379)], for 5 min at RT. Subsequently, endogenous peroxidase activity was blocked by 0.3% $H_2O_2$/PBS (Merck, 107210.0250) for 10 min in the dark at 4°C. Slides were washed three times in TBT for 5 min at RT.

Non-specific binding was blocked with Super Block (ScyTek, AAA999) incubation in humid chamber for 10 min at RT, followed by incubation with the first antibody SOX9 (Millipore, AB5535, 1:8000), with overnight incubation in a humid chamber at 4°C. The next day, the slides were washed three times in TBT for 5 min at RT. Slides were incubated with BrightVision polyHRP Ms+Rb (Immunologic, VWRKDPVO110HRP) in a humid chamber for 10 min at RT. Subsequently, the slides were incubated with Opal 480 [Akoya, FP1500001KT, 1:100 in amplification diluent (Akoya, FP1498)] for 10 min in the dark in a humid chamber for 10 min at RT. Slides were washed three times in TBT for 5 min at RT. Antigen retrieval were performed in a microwave until boiling 100°C for 15 min with citrate buffer [0.01 M sodium citrate dihydrate (Roth, 4088.1) (pH 6.0)]. Slides were cooled down before proceeding with the protocol. Slides were washed three times in TBT, for 5 min at RT. For the following antibodies this process was repeated, in this case AMH (Santa Cruz Biotechnology, SC-166752, 1:500) coupled with Opal 570 (Akoya, FP1488001KT, 1:100 in amplification diluent); AR coupled with Opal 690 (Akoya, FP1495001KT, 1:100 in amplification diluent). For visualisation, slides were embedded with Prolong Gold Antifade Reagent (Thermo Fisher Scientific, P36930) and were stored several hours to dry at 4°C.

#### Multiplex tyramine staining procedure of germ cell maturation and proliferation
For a list of antibodies, see Table S7. Testicular tissue sections were deparaffinised in xylene I and xylene II, each for 5 min, followed by hydration of the tissue sections in a decreasing concentration of EtOH (100% EtOH I, 100% EtOH II, 96% EtOH and 70% EtOH), each for 2 min. Post tissue fixation were performed with 4% formaldehyde for 20 min at RT. Slides were washed three times in TBT, for 5 min at RT. Subsequently, sections were blocked with freshly prepared endogenous peroxidase 0.3% $H_2O_2$/PBS for 10 min in the dark. Slides were washed three times in TBT for 5 min at RT. Non-specific binding was blocked with Super Block incubation in a humid chamber for 10 min at RT, followed by incubation with the first antibody MAGE-A4 (Millipore, MABC11150, 1:10.000) with overnight incubation in a humid chamber at 4°C. The next day, slides were washed three times in TBT for 5 min at RT. Slides were incubated with BrightVision polyHRP Ms+Rb in a humid chamber for 10 min at RT. Subsequently, slides were incubated with Opal 520 (Akoya, FP1494001KT, 1:100 in amplification diluent) for 10 min in the dark in a humid chamber for 10 min at RT. Antigen retrieval was performed in a microwave until boiling 100°C for 15 min with tris-ethylenediaminetetraacetic acid (EDTA) buffer [10 mM Tris and hydrochloric acid (Merck, 1003171000), 1 mM disodium EDTA (Merck, 108,418,250) (pH 9)]. Slides were cooled down before proceeding with the protocol. Slides were washed three times in TBT for 5 min at RT. For the following antibodies, this process was repeated, in this case SOX9 (Millipore, AB5535, 1:8000) coupled with Opal 570 (1:100 in amplification diluent); Ki67 (Dako, M7240, 1:500) coupled with Opal 690 (1:100 in amplification diluent); OCT3/4 (Santa Cruz Biotechnology, A1723, 1:1000) coupled with TSA-DIG (Akoya, NEL748001KT, 1:100) and Opal 780 (Akoya, FP1495001KT, 1:25 in amplification diluent). For visualisation, slides were embedded with Prolong Gold Antifade Reagent and were stored several hours to dry at 4°C.

#### Data visualisation
Images for manual cell counting were acquired using Leica bright field and Phase Contrast Microscope DM6. Images of testicular tissue sections were captured at 20× magnification and visualised using LAS X Life Science Microscope Software (LASX Office 1.4.5 27713).

Imaging for automatic cell counts was performed with Confocal Stellaris 8 (without using confocal options or z planes), for high-quality imaging. Images of whole testicular tissue sections were captured at 20× magnification and visualised using LAS X Life Science Microscope Software and analysed in QuPath 0.5.1. All post imaging settings were defined for each immunofluorescent run after observing the (co)expression of markers in positive and negative control samples.

### Quantifications and statistical analysis

#### Measurement of tubules diameter
Quantitative assessment of seminiferous tubule diameter was performed to evaluate potential maturation alterations. All identifiable tubules within each

histological section were systematically measured, and the mean diameter was calculated for each individual sample. To ensure methodological consistency and minimise variability due to sectioning or tissue deformation, measurements were restricted to the shortest cross-sectional axis of each tubule. To evaluate the proportional distribution between the tubules and interstitial compartments, the total area of each testicular tissue section, along with the cumulative area occupied by the seminiferous tubules, was determined and quantified using QuPath image analysis software. The interstitial area was subsequently determined by subtracting the tubular area from the total tissue area.

## Positive cell quantification

Immunofluorescent multiplex staining was performed to observe the marker expression in Sertoli and germ cells. General morphology and staining patterns were assessed. All identifiable tubules within each histological section were systematically measured for each individual sample. All tubules in the testicular tissue section were included for analysis. Tissue sections were divided into the intratubular area and interstitial space (unless no tubule formation was visible). The amount of tubules was determined, and general morphology and staining patterns were assessed. The presence of immunofluorescent signal within the intratubular area was analysed as single-positive or double-positive signal. Any remarkable features of the tissue were documented separately.

## Sertoli cell dynamics: SOX9-AMH-AR

AMH quantification was performed by analysing intratubular area classified based on expression intensity. If more than 50% of the tubule showed bright AMH expression, the intratubular area was classified as positive for AMH; otherwise, they were classified as negative. Sertoli cell AR expression was defined as expression of AR in SOX9+ cells. Brightness of AR expression was compared relative to that in peritubular cells, to determine positive expression of AR within the intratubular area. If more than 50% of the tubules showed bright AR expression, they were classified as positive; otherwise, they were classified as negative.

## Sertoli-germ cell interaction: MAGE-A4, OCT3/4, Ki67 and SOX9

Quadruple colorimetric immunofluorescent staining was performed to understand the dynamics between Sertoli and germ cells. Signal intensity settings were confirmed by adjusting the background intensity. Using QuPath, SOX9 area and cell count were acquired automatically; all other cell counts were performed manually. Analysis was performed in tubule areas with positive SOX9 expression with and without Ki67 and in the same tubule for germ cells with positive expression for OCT3/4 and/or MAGE-A4 with or without Ki67. Raw data and counts were extracted from QuPath.

## Statistical analysis

For data analysis, calculations and visualisation, GraphPad Prism version 10.2.0 was used to process the raw counting results. This included the generation of various graphical outputs such as descriptive statistical analysis, bar charts, plots, and other visualisations. This visualisations tool facilitated a comprehensive evaluation of the results by highlighting patterns, differences between groups, and trends over time.

## Acknowledgements

We are grateful to the donors and their families. We also thank the Human Developmental Biology Resource (www.hdbr.org) for providing some of the foetal material (Joint MRC/Wellcome Trust; 099175/Z/12/Z). We also extend our gratitude to the staff of the core facility Cellular Imaging of the Advanced Light Microscopy Centre Amsterdam, especially Diasy Picavet, Amsterdam UMC, for technical support.

## Competing interests

The authors declare no competing or financial interests.

## Author contributions

Conceptualization: I.S., M.L., R.T.M., C.L.M.; Data curation: I.S., M.L.; Formal analysis: I.S., M.L.; Investigation: I.S., M.L.,; Methodology: I.S., M.L., L.K.S., S.K.M.v.D., C.M.d.W.-K.; Resources: A.M., R.T.M.; Supervision: G.H., R.T.M., C.L.M.; Visualization: I.S., M.L., L.K.S., S.K.M.v.D., C.M.d.W.-K.; Writing – original draft: I.S., M.L., C.L.M.; Writing – review & editing: I.S., M.L., L.K.S., S.K.M.v.D., C.M.d.W.-K., A.M., G.H., D.G.d.R., R.T.M., C.L.M.

## Diversity and inclusion

This study was conducted with a commitment to diversity, equity, and inclusion. We fostered an inclusive environment that supports diverse scientific perspectives. Our work advances understanding of developmental processes critical to male reproductive biology, with potential implications for diverse populations at risk of infertility.

## Funding

No specific funding was acquired for this study. R.T.M. is supported by a UK Research and Innovation (UKRI) Future Leaders Fellowship (MR/Y011783/1). Open Access funding provided by University of Amsterdam. Deposited in PMC for immediate release.

## Data and resource availability

All relevant data and details of resources can be found within the article and its supplementary information. Details of software tools can be found in the Materials and Methods section.

## First Person

This article has an associated First Person interview with the joint first authors of the paper.

## Peer review history

The peer review history is available online at https://journals.biologists.com/bio/lookup/doi/10.1242/bio.062319.reviewer-comments.pdf

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
