## [Peer Review File · Biology Open]

Temporal dynamics of Sertoli and germ cell development in human foetal and prepubertal testis

Iris Sanou, Mathangi Lakshmipathi, Lisette K. Schönhage, Saskia K. M. van Daalen, Cindy M. de Winter-Korver, Andreas Meißner, Geert Hamer, Dirk G. de Rooij, Rod T. Mitchell and Callista L. Mulder

DOI: 10.1242/bio.062319

Editor: Tristan Rodriguez

Review timeline

Original submission:	14 October 2025
Editorial decision:	23 October 2025
First revision received:	30 December 2025
Accepted:	5 January 2025

Original submission

First decision letter

MS ID#: bio. 062319

MS Title: Temporal dynamics of Sertoli and germ cell development in human foetal and prepubertal testis

Authors: Iris Sanou; Mathangi Lakshmipathi; Lisette K. Schönhage; Saskia K. M. van Daalen; Cindy M. de Winter-Korver; Andreas Meißner; Geert Hamer; Dirk G. de Rooij; Rod T. Mitchell; Callista L. Mulder

Article Type: Research Article

Dear Dr Mulder,

I have now reached a decision on the above manuscript.

The reviewer reports are shown at the bottom of this email or can be accessed, together with a copy of this decision letter, by going to:

As you will see, the reviewers gave favourable reports, but raised some critical points that will require amendments to your manuscript. I hope that you will be able to carry these out, because we would like to be able to accept your paper.

Reviewer 1

Comments for the author

This is an interesting and valuable work, as it collects and analyses in detail a collection of precious fetal and pre-pubertal testicular samples from human subjects. It is rare to have access and study together, using standardized assays and technical approaches, to such material. Furthermore, the authors have done a good job in studying these samples, where major potential criticisms would come from a lack of more samples, what is clearly not easy at all. This reviewer only finds minor issues with the manuscript, detailed below.

1. Abbreviated gene names with no description should be avoided in the abstract.
2. In the first paragraph of the results, different cell types are mentioned (germ cells, Sertoli cells, peritubular myoid cells, other somatic cells), but only an expert eye would be able to distinguish them in the H&E staining shown in Fig. 1A.
3. What are the arrowheads shown in the different panels of Fig 1A pointing at?
4. Lines 120-122: "The number and types of developing germ cells present also increased with age, with spermatocytes and spermatids present around 10 and 13.5 years old respectively." What and where is the evidence to support this statement?
5. The number of samples for each time point used should be indicated in Fig. 1B-E, even though the information is available in supplementary Table S6. Despite the inevitable low number of samples per time point, the trends shown in Fig. 1B, C, and E is clear. However, the variability in Fig. 1D could well be due to this, so it is not that obvious to state "In contrast, prepubertal testes displayed greater variability in the relative proportions of tubules and interstitial areas" (lines 114-115) as the authors do.
6. The data shown in supplementary Table S2 is very poorly, if at all, described.
7. The age of the samples shown in Figs 2B, 3B and 4A-C should be indicated, in both the figure and the legend. Although certainly this information can be found in the text of the Results section, it would help the readers to find it as well in the figures.
8. Lines 188-189: "Compared to OCT3/4+ gonocytes, these cells tend to be larger, with rounder nuclei and typically localised at the basal side." Where is the evidence to support this statement? It is difficult to appreciate these differences, as well as other described later (lines 208-212) from the images shown in Fig. 3A.
9. Line 214: the authors cannot state that "... most proliferating intratubular cells co-expressed Ki67 and OCT3/4", if just following they explain that the percentage of coexpressing cells ranges from 0 to 40%, with quite high variability.

Reviewer 2

Comments for the author

Using human foetal and prepubetal testicular samples, Sanou et al., investigated the temporal dynamics of Sertoli and germ cell maturation by characterizing their developmental and proliferative states. The authors demonstrate that loss of anti-Müllerian hormone (AMH), gain of androgen receptor (AR), and lumen formation mark the transition from proliferative to mature Sertoli cells, indicating somatic testicular maturation occurs before the puberty (from approximately 4 yrs of age). This suggests that Sertoli cell numbers are established early in childhood. Additionally, OCT3/4-positive germ cells are proliferative during fetal development, whereas MAGE-A4-expressing pre-spermatogonia divide slowly until puberty onset (around 13.5 years). These findings provide valuable reference data for understanding testis maturation with potential application in fertility preservation research.

That said, while acknowledging challenges associated with obtaining such rare human samples, the experimental methodology is inadequately designed and/or insufficiently described. The manuscript requires thorough English language and copy editing, as well as substantial improvements in clarity, organization, reproducibility, and overall readability. Specific suggestions are provided below but are not exhaustive:

1. Writing and presentation: The authors should carefully revise the manuscript for English usage, grammar, and organization, as many sections are written too vaguely. For example:

a. Methodology example (Lines 380-390): "Testicular tissue sections were deparaffinise in xylene, rehydrating in a decreasing concentration of ethanol. Post fixation is performed with 4% formaldehyde. Slides were washed in TBT (Tris-Buffered Saline, 0.05% Tween-20)(Sigma, P1379). Subsequently, sections were blocked with endogenous peroxidase 0,3% H₂O₂/PBS (Merck, 107210.0250). Non-specific binding was blocked with SuperBlock, followed by incubation with the first antibody with overnight incubation. Next day, incubation with secondary antibody was performed. After washing, slides were incubated with opal fluorophore. After incubation antigen retrieval was performed in a microwave with citrate buffer (pH 6). This procedure was repeated for all first antibodies against specific markers of interest, using another fluorophore to detect each marker and its overlay with other markers. Finally, the slides were mounted with Prolong Gold Antifade Reagent"

In addition to grammatical issues - deparaffinise, rehydrating, post fixation is, 0,3% H₂O₂, etc, to improve clarity and reproducibility, the following details should be included or clarified:

- * Specify the concentrations of ethanol used during dehydration.
- * Provide the duration and conditions-temperature, buffer composition, for blocking.
- * Include detailed information on secondary antibody-source, catalog number, dilution.
- * Specify the incubation times, dilutions, and conditions-temperature, agitation for the primary, secondary and Opal fluorophore steps.
- * Mention the duration and temperature of antigen retrieval.
- * The multiIF sequentially staining method is overly vague and difficult to reproduce. The authors should either clearly describe all steps or cite the previously validated protocols.
- * Although antibody dilutions and reagent details are mentioned in the Supplemental data, these are incomplete. It is strongly recommended to incorporate this information into the methods section.
- * Importantly, multi-IF was cited only as an example, however all protocols described in the method section require similar attention to details to improve clarity and reproducibility.

b. Figures and legends: The data presented for lumen formation, Sertoli cell, and germ cell maturation markers in figure 1 to 3 correspond to different time points, making cross-comparison difficult. Authors should consider including the additional intermediate time points in the supplemental information.

- * Figures with quantitative data are presented without statistical analyses. Statistical tests and corresponding p-values should be reported.
- * The number of samples and biological replicates for each time point is unclear and must be explicitly stated in each figure legend.
- * Figure legends should include sufficient details, such as what error bar represent-SD, SE, number of samples, and statistical tests.
- * Some graphs show error bars only for selected data points. Does that indicate limited sample power? Please clarify.
- * Label different time point montages clearly and refer specific timepoint/label in the text for easy interpretation, rather than Fig 1A, 2A, and 3A.
- * Arrowheads and asterisks in figures are barely visible, ensure adequate contrast and specify their meaning in the legend. Recheck the magenta and yellow arrowheads in Figure 2A.
- * Indicate fetal time PCW for Figures 2B, 3B, and 4A.
- * Include a high-resolution insert for Figure 1A to better visualize lumen formation across different PCWs.
- * Consider adding an image showing SOX9 and AMH co-localization in early PCW.
- * In figure 4A, immunofluorescence images appear to multiple fluorophores, whereas the describes only two markers. Please clarify this discrepancy.
- * A schematic summarizing the developmental timeline-processes, gene expression patterns, and a diagram showing tubules and interstitial regions would greatly aid readers.
- * Line 107: "Later in development around PCW12....." - the corresponding Figure in 1A labels PCW 13.
- * Line 134: "From approximately PCW 14 onwards....." - data in the corresponding Figure 2A mentions PCW 12.

c. Rather than reiterating study findings, the Discussion should place the results in context with existing literature on primate and rodent testicular development. The authors should also discuss

study limitations, including discuss the limitations including sample number, inter-sample variability, and potential implications for interpretation.

2. Methodology: As stated above, the authors must clearly specify sample numbers, replicates, and p-values for all quantitative data.

* Several claims are made without supporting data. For instance, Lines 121-122: "... with spermatocytes and spermatids present around 10 and 13.5 years old respectively"; lines 163-164: "...AR was not detected in Sertoli cells during early postnatal period but began to increase gradually from around 4 years of age....".

If these statements refer to supplemental data, cite them explicitly in the text.

* Lines 136-137: Clarify how did germ cells were visualized. Were all six reported markers applied on the same tissue sections in multi-IF staining?

* Line 187: MAGE-A4 expression is shown at PCW 11 in Figure 3A, but text mention PCW 9. In Figure 3C quantification shows variable expression across different fetal PCWs. Please clarify.

* Line 198-199: The text states a "gradual increase in the number of both OCT3/4+ gonocytes and MAGE-A4+ pre-spermatogonia per tubular cross section with advancing gestational age." However, Figure 3C shows the OCT3/4 increasing between PCW 8 to 10 and MAGE-A4 after PCW14. Given this variability, clarify whether the differences are statistically significant.

* State the rationale clearly before reporting the experimental observations in the result section.

Minor edits:

1. Avoid abbreviations in the abstract.
2. Lines 42: "Male reproductive capacity....." Specify human/primate.
3. Line 93-97: The paragraph end abruptly-consider summarizing the study outcomes and implications.
4. Line 52; Expand "SRY"
5. Line 126: Remove "%"
6. Line 169: Replace "LH" with "Luteinizing Hormone"
7. Line 338: Change "n=14, 2-13,5 years old" to ""n=14, 2-13.5 years old"
8. Line 396: Clarify whether the 8 images correspond to the same section or Z-plane, and indicate section depth if applicable.
9. Line 712: Replace "Haematoxylin" with "Hematoxylin"
10. Line 712 and 714: Replace "HE" with "H&E"
11. Line 715: Replace "6,5" with "6.5"
12. Replace "Qupath" with "QuPath"
13. Include Y-axis units for Figures 2B and D

Reviewer's Responses to Questions

Experimental quality

Does each figure have the proper controls?

If 'No', please indicate reasons in Comments for Author box below.

Reviewer #1:

- Yes

Reviewer #2:

- Yes

Were the data analyzed using appropriate statistical tests?

If 'No', please indicate reasons in Comments for Author box below.

Reviewer #1:

- Yes

Reviewer #2:

- No

Reproducibility

Were experiments performed using adequate number of biological replicates?

If 'No', please indicate reasons in Comments for Author box below.

Reviewer #1:

- Yes

Reviewer #2:

- No

Does the methods section provide sufficient detail to permit reproducibility?

If 'No', please indicate reasons in Comments for Author box below.

Reviewer #1:

- Yes

Reviewer #2:

- No

Completeness

Are the manuscript's conclusions supported by the data?

If 'No', please indicate reasons in Comments for Author box below.

Reviewer #1:

- Yes

Reviewer #2:

- No

Scholarship

Do the authors cite and discuss the merits of data that would argue for and against their conclusion?

If 'No', please indicate reasons in Comments for Author box below.

Reviewer #1:

- Yes

Reviewer #2:

- Yes

Does the manuscript title & abstract accurately reflect the contents of the manuscript, without hyperbole?

If 'No', please indicate reasons in Comments for Author box below.

Reviewer #1:

- Yes

Reviewer #2:

- No

First revision

Author response to reviewers' comments

All changes to the manuscript are indicated in blue and underlined in the MARKED version of the manuscript.

Answers to Reviewer: 1

Comments from the Reviewers:

Reviewer 1: This is an interesting and valuable work, as it collects and analyses in detail a collection of precious fetal and pre-puberal testicular samples from human subjects. It is rare to have access and study together, using standardized assays and technical approaches, to such material. Furthermore, the authors have done a good job in studying these samples, where major potential criticisms would come from a lack of more samples, what is clearly not easy at all. This reviewer only finds minor issues with the manuscript, detailed below.

Thank you for your comprehensive review of our manuscript. We appreciate your time and effort in critically evaluating our work to help enhance its quality. Below, we have provided detailed responses to your questions and suggestions, with references to the updated line numbers in the revised MARKED version of the manuscript, where all changes are indicated in blue and are underlined.

1. Abbreviated gene names with no description should be avoided in the abstract.

Answer: We have now included the expanded names of the factors - “*anti-Müllerian hormone (AMH)*”, “*Androgen Receptor (AR)*”, “*Octamer binding transcription factor (OCT) 3/4*” and “*Melanoma associated antigen (MAGE)-A4*” in L32-37 of the abstract.

2. In the first paragraph of the results, different cell types are mentioned (germ cells, Sertoli cells, peritubular myoid cells, other somatic cells), but only an expert eye would be able to distinguish them in the H&E staining shown in Fig. 1A.

Answer: We thank the reviewer for highlighting this. We have added arrows, empty and closed arrowheads to pinpoint the specific cell types and included peritubular myoid cells. ‘*Closed arrowheads indicate germ cells, open arrowheads indicate Sertoli cells and arrows indicate PTM cells*’ in L844-845.

3. What are the arrowheads shown in the different panels of Fig 1A pointing at?

Answer: We have included this information in the figure legend. The closed arrowhead indicates germ cells, open arrowhead indicates Sertoli cells and arrows indicates peritubular myoid cells.

4. Lines 120-122: “The number and types of developing germ cells present also increased with age, with spermatocytes and spermatids present around 10 and 13.5 years old respectively.” What and where is the evidence to support this statement?

Answer: We thank the reviewer for this comment. To clarify, we have revised the sentence to - “*The range of developing germ cell types observed increased with age, with spermatocytes and spermatids appearing around 10 and 13.5 years old respectively (Supplementary table S2).*” in L138-141. To support this, we have also added a column indicating the most advanced germ cell type observed in each prepubertal sample in Supplementary table S2.

5. The number of samples for each time point used should be indicated in Fig. 1B-E, even though the information is available in supplementary Table S6. Despite the inevitable low number of samples per time point, the trends shown in Fig. 1B, C, and E is clear. However, the variability in Fig. 1D could well be due to this, so it is not that obvious to state “In contrast, prepubertal testes displayed greater variability in the relative proportions of tubules and interstitial areas” (lines 114-115) as the authors do.

Answer: We thank the reviewer for this observation. We have now rephrased this sentence to acknowledge this limitation as follows - *“In contrast, prepubertal testes displayed greater variability in the relative proportions of seminiferous compartment and interstitial areas, although this may in part be attributable to the low number of biological replicates available for these samples.”* in L130-132.

In addition, we have also included the number of biological replicates for each time point in the figure legends of Fig 1, 2, 3 and 4.

6. The data shown in supplementary Table S2 is very poorly, if at all, described.

Answer: We thank the reviewer for this comment. We have added additional data in this table (now term supplementary table S3) “Sertoli cell maturation”. This is also described in L537-544 in the methods section - *‘AMH quantification was performed by analysing intratubular area classified based on expression intensity. If more than 50% of the tubule showed bright AMH expression, intratubular area was classified as positive for AMH; otherwise, they were classified as negative. Sertoli cell AR expression was defined as expression of AR in SOX9 positive cells. Brightness of AR expression was compared relative to peritubular cells, to determine positive expression of AR within the intratubular area. If more than 50% of the tubules showed bright AR expression, they were classified as positive; otherwise, they were classified as negative.’*

7. The age of the samples shown in Figs 2B, 3B and 4A-C should be indicated, in both the figure and the legend. Although certainly this information can be found in the text of the Results section, it would help the readers to find it as well in the figures.

Answer: To indicate the age of the higher magnification images of co-expression, we have rephrased this section in the figure legends accordingly:

Fig 2B: *“Higher-magnification image from a PCW 20 foetal testis showing SOX9 (in magenta) and AR (in cyan) co-expression, with white arrowheads indicating SOX9 and AR double-positive Sertoli cells. Scale bar = 100 μ m.”*

Fig 3B: *“Higher-magnification image from a PCW 10 foetal testis showing SOX9 (green), OCT3/4 (red), and MAGE-A4 (blue), with white arrowheads indicating OCT3/4 and MAGE-A4 double-positive germ cells. Scale bar = 100 μ m.”*

Fig 4A-C: *“A. Representative image showing co-staining of OCT3/4 (red) and Ki67 (yellow) from a PCW 14 foetal testis, with white arrowheads indicating double-positive cells. Scale bar = 50 μ m B. Representative image showing co-staining of MAGE-A4 (blue) and Ki67 (yellow) from a prepubertal testis aged 8.5 years old, with white arrowheads indicating double positive cells. Scale bar = 50 μ m C. Representative image showing co-staining of SOX9 (green) and Ki67 (yellow) from a PCW 16 foetal testis, with white arrowheads indicating double positive cells. Scale bar = 50 μ m.”*

8. Lines 188-189: “Compared to OCT3/4+ gonocytes, these cells tend to be larger, with rounder nuclei and typically localised at the basal side.” Where is the evidence to support this statement? It is difficult to appreciate these differences, as well as other described later (lines 208-212) from the images shown in Fig. 3A.

Answer: We thank the reviewer for this comment. To clarify this and provide visual evidence supporting these observations, we have now included *Supplementary figures S2A and S2B*.

Fig S2A shows foetal samples (PCW 12, 14 and 19) stained for OCT3/4 (in red), MAGE-A4 (in blue) and SOX9 (in green). These images illustrate that OCT3/4+ gonocytes possess smaller nuclei and are primarily located in luminal regions, while MAGE-A4 prespermatogonia display relatively larger nuclei and are positioned closer to the basal membrane.

Fig S2B includes prepubertal samples (2.5years and 10.5 years), highlighting that MAGE-A4+ spermatogonia are occasionally observed at adluminal positions at younger ages (2.5y), but become exclusively restricted to the basal compartment from at 10.5y.

We refer to this figure in L314-214, L225-229 and L236-239.

9. Line 214: the authors cannot state that "... most proliferating intratubular cells co-expressed Ki67 and OCT3/4", if just following they explain that the percentage of coexpressing cells ranges from 0 to 40%, with quite high variability.

Answer: We thank the reviewer for this comment. To clarify this, we have changed the y-axis in figure 4 to: "**Percentage of Ki-67+ cells per cell population**". In the main text of the manuscript, we state that: "*The percentage of proliferative cells within the OCT3/4-positive cell population, ranged between 4% and 41% across samples, varying considerably between samples (Fig. 4 A, D and Supplementary table S4), in line with previous reports (Gaskell et al., 2004, Pauls et al., 2006, Saxe et al., 2009).*" in L244-247. We also changed this in the figure legends: to "**D. Percentage of proliferative Ki-67+ cells within the SOX9, OCT3/4 and MAGE-A4 cell population.**"

Answers to Reviewer: 2:

Reviewer 2: Using human foetal and prepubertal testicular samples, Sanou et al., investigated the temporal dynamics of Sertoli and germ cell maturation by characterizing their developmental and proliferative states. The authors demonstrate that loss of anti-Müllerian hormone (AMH), gain of androgen receptor (AR), and lumen formation mark the transition from proliferative to mature Sertoli cells, indicating somatic testicular maturation occurs before the puberty (from approximately 4 yrs of age). This suggests that Sertoli cell numbers are established early in childhood. Additionally, OCT3/4-positive germ cells are proliferative during fetal development, whereas MAGE-A4-expressing pre-spermatogonia divide slowly until puberty onset (around 13.5 years). These findings provide valuable reference data for understanding testis maturation with potential application in fertility preservation research.

That said, while acknowledging challenges associated with obtaining such rare human samples, the experimental methodology is inadequately designed and/or insufficiently described. The manuscript requires thorough English language and copy editing, as well as substantial improvements in clarity, organization, reproducibility, and overall readability. Specific suggestions are provided below but are not exhaustive:

Thank you for your comprehensive review of our manuscript. We appreciate your time and effort in critically evaluating our work to help enhance its quality. Below, we have provided detailed responses to your questions and suggestions, with references to the updated line numbers in the revised **MARKED** version of the manuscript, where all changes are indicated in blue and are underlined.

1. Writing and presentation: The authors should carefully revise the manuscript for English usage, grammar, and organization, as many sections are written too vaguely. For example:

a. Methodology example (Lines 380-390): "Testicular tissue sections were deparaffinise in xylene, rehydrating in a decreasing concentration of ethanol. Post fixation is performed with 4% formaldehyde. Slides were washed in TBT (Tris-Buffered Saline, 0.05% Tween-20)(Sigma, P1379). Subsequently, sections were blocked with endogenous peroxidase 0,3% H2O2/PBS (Merck, 107210.0250). Non-specific binding was blocked with SuperBlock, followed by incubation with the first antibody with overnight incubation. Next day, incubation with secondary antibody was performed. After washing, slides were incubated with opal fluorophore. After incubation antigen retrieval was performed in a microwave with citrate buffer (pH 6). This procedure was repeated for all first antibodies against specific markers of interest, using another fluorophore to detect each marker and its overlay with other markers. Finally, the slides were mounted with Prolong Gold Antifade Reagent"

In addition to grammatical issues - deparaffinise, rehydrating, post fixation is, 0,3% H2O2, etc, to improve clarity and reproducibility, the following details should be included or clarified:

Answer: Thank you for your comprehensive review of our manuscript. We apologize for our English grammar and sentence organisation. For this reason, we have made a significant number of changes within this section.

* Specify the concentrations of ethanol used during dehydration.

Answer: We thank the reviewer for this comment. To clarify this, we rephrased to '*Testicular tissue sections were deparaffinised in xylene I and xylene II, each for 5 minutes, followed by hydration of the tissue sections in a decreasing concentration of ethanol (100% EtOH I, 100% EtOH II, 96% EtOH and 70% EtOH), each for 2 minutes at room temperature.*' in L457-460.

* Provide the duration and conditions-temperature, buffer composition, for blocking.

Answer: We thank the reviewer for this comment. Super Block is an off the shelf and ready-to-use product. We have changed the lines into '*Non-specific binding was blocked with Super Block (ScyTek, AAA999) incubation in humid chamber for 10 minutes at RT,....*' in L465-467.

* Include detailed information on secondary antibody-source, catalog number, dilution.

Answer: We thank the reviewer for this comment. Previously we mentioned the secondary antibody information only in the supplementary table S7. We now incorporated this information in the text in the method section in L457.

* Specify the incubation times, dilutions, and conditions-temperature, agitation for the primary, secondary and Opal fluorophore steps.

Answer: We thank the reviewer for this comment. We have added the incubation times, dilutions, temperature, primary antibodies and Opal fluorophore steps in the section of Multiplex Tyramine Staining procedure in L455-502.

* Mention the duration and temperature of antigen retrieval.

Answer: We thank the reviewer for this comment. We have added the duration and temperature of antigen retrieval in L472-473 and L492-495.

* The multiIF sequentially staining method is overly vague and difficult to reproduce. The authors should either clearly describe all steps or cite the previously validated protocols.

Answer: We have made a significant number of changes within this section to enhance readability and reproducibility in L444-502.

* Although antibody dilutions and reagent details are mentioned in the Supplemental data, these are incomplete. It is strongly recommended to incorporate this information into the methods section.

Answer: We thank the reviewer for this comment. We have added the antibody incubation times, dilutions, and conditions-temperature in a section on the Multiplex Tyramine Staining procedure in L444-501 and supplementary table S7.

* Importantly, multi-IF was cited only as an example, however all protocols described in the method section require similar attention to details to improve clarity and reproducibility.

Answer: We have made a significant number of changes within this section in L444-501.

b. Figures and legends: The data presented for lumen formation, Sertoli cell, and germ cell maturation markers in figure 1 to 3 correspond to different time points, making cross-comparison difficult. Authors should consider including the additional intermediate time points in the supplemental information.

Answer: We thank the reviewer for this suggestion. In the main figures of the manuscript, we selected representative developmental ages that best illustrate the specific claim discussed in the manuscript. However, to facilitate better cross-comparison and to provide a more continuous developmental overview, we have now included the intermediate ages in *Supplementary figure S3*.

This figure presents prepubertal testis samples spanning 2.5- 10 years. The top row indicates Sertoli cell maturation markers SOX9 (in magenta), AMH (in yellow) and AR (in cyan), while the bottom row displays germ cell markers OCT3/4 (in red) and MAGE-A4 (in blue), together with SOX9 (in green) to delineate the seminiferous tubules. As expected, OCT3/4+ gonocytes are absent in post-natal samples due to the loss of the remaining pluripotent germ cells in infancy.

* Figures with quantitative data are presented without statistical analyses. Statistical tests and corresponding p-values should be reported.

Answer: We appreciate the reviewer's suggestion to include statistical analyses. In the current study, the number of independent experiments for some timepoints is low (in certain cases 1 or 2) due to the scarcity of tissue. Under these conditions, conventional statistical testing would not be meaningful and could be potentially misleading. Instead, we have now included a supplementary table (Supplementary table S4) presenting descriptive statistics (N number, mean and standard deviation) to allow transparency of the data, which is referred to in L150-247. We believe this approach provides a fair and informative representation of the data, given the experimental constraints.

* The number of samples and biological replicates for each time point is unclear and must be explicitly stated in each figure legend.

Answer: We thank the reviewer for this comment. We have now incorporated the number of samples and biological replicates as : “*n=3 for PCW 11, 12 and 16-20; n=2 for PCW 7, 9, 10, 13 and 14, 4y, 6.5y and 13.5y ; n=1 for PCW 8, 15 and 21, 2y, 2.5y, 7y, 8y, 8.5y, 10y, 10.5y and 11y*” in all the figure legends.

* Figure legends should include sufficient details, such as what error bar represent-SD, SE, number of samples, and statistical tests.

Answer: To make this clear, we have included in the figure legends that the “*Data shown are mean \pm s.d.*”, along with the number of biological replicates for each time point.

* Some graphs show error bars only for selected data points. Does that indicate limited sample power? Please clarify.

Answer: We thank the reviewer for this question. The absence of error bar for some data points indeed reflects limited biological replicates for those time points. As previously mentioned, to clarify this, we have now included the number of biological replicates per time point in the corresponding figure legends.

* Label different time point montages clearly and refer specific timepoint/label in the text for easy interpretation, rather than Fig 1A, 2A, and 3A.

Answer: We thank the reviewer for this comment. We have now included specific time points to the figure citations in the manuscript, wherever relevant.

* Arrowheads and asterisks in figures are barely visible, ensure adequate contrast and specify their meaning in the legend. Recheck the magenta and yellow arrowheads in Figure 2A.

Answer: To improve the indications in figure 1A and 2A, we have made the following changes to the figures and have indicated their meaning in the respective legends:

Fig 1A: We thank the reviewer for highlighting this. We have now enlarged the size of the arrowheads and arrows for clarification of the various cell types. We have also included peritubular myoid cells in figure 1 and explained this in the figure legend.

Fig 2A: We have now enlarged the size of the arrowheads in the figure to make the co-localisation of SOX9 and AMH more clear. In addition, we have included the sentence - “Magenta and yellow arrowheads indicate the polarisation of Sertoli cells, marked by SOX9 and AMH respectively, during this developmental window.” to the legend of Fig 2A.

* Indicate fetal time PCW for Figures 2B, 3B, and 4A.

Answer: To indicate the age of the higher magnification images of co-expression, we have added the ages in the figures and rephrased this section in the figure legends accordingly:

Fig 2B: “Higher-magnification image from a PCW 20 foetal testis showing SOX9 (in magenta) and AR (in cyan) co-expression, with white arrowheads indicating SOX9 and AR double-positive Sertoli cells. Scale bar = 100 μ m.”

Fig 3B: “Higher-magnification image from a PCW 10 foetal testis showing SOX9 (green), OCT3/4 (red), and MAGE-A4 (blue), with white arrowheads indicating OCT3/4 and MAGE-A4 double-positive germ cells. Scale bar = 100 μ m.”

Fig 4A: “A. Representative image showing co-staining of OCT3/4 (red) and Ki67 (yellow) from a PCW 14 foetal testis, with white arrowheads indicating double-positive cells. Scale bar = 50 μ m B. Representative image showing co-staining of MAGE-A4 (blue) and Ki67 (yellow) from a prepubertal testis aged 8.5 years old, with white arrowheads indicating double positive cells. Scale bar = 50 μ m C. Representative image showing co-staining of SOX9 (green) and Ki67 (yellow) from a PCW 16 foetal testis, with white arrowheads indicating double positive cells. Scale bar = 50 μ m.”

* Include a high-resolution insert for Figure 1A to better visualize lumen formation across different PCWs.

Answer: We would like to clarify to the reviewer that lumen formation is a hallmark of Sertoli cell maturation, and therefore, is not observed during foetal life. In our sample cohort, lumen formation was first observed around 6.5 years of age and became more prominent by approximately 8 years of age, as described in the manuscript in L134-136 and in supplementary figure S3.

* Consider adding an image showing SOX9 and AMH co-localization in early PCW.

Answer: We appreciate the reviewer’s suggestion. To address this, we have included Supplementary figure S1A, showing foetal testis tissue at PCW 7 stained for SOX9 (in magenta) and AMH (in yellow). The merged and individual channels clearly demonstrate co-localisation of SOX9 and AMH during this early developmental stage.

* In figure 4A, immunofluorescence images appear to multiple fluorophores, whereas the describes only two markers. Please clarify this discrepancy.

Answer: We thank the reviewer for pointing this out. In Figure 4A, multiple fluorophores are visible because the images include additional channels used during imaging. However, for clarity and to specifically highlight the co-staining of Ki67 with the respective marker, only these two markers are shown and quantified in the results. This is presented in Figure 4D.

* A schematic summarizing the developmental timeline-processes, gene expression patterns, and a diagram showing tubules and interstitial regions would greatly aid readers.

Answer: We thank the reviewer for this valuable suggestion. In response, we have now included Figure 5, which summarises the key findings of our study. This schematic integrates the developmental timeline with major milestones in testicular development, alongside representative gene expression patterns. Additionally, we have included a new subsection in the Results titled “A timeline of maturation dynamics in the human testis” in L266-276 which consolidates and contextualises all relevant findings.

* Line 107: "Later in development around PCW12...." - the corresponding Figure in 1A labels PCW 13.

Answer: We thank the reviewer for noticing this discrepancy. To avoid the confusion, we have now removed the sentence in L121-123.

* Line 134: "From approximately PCW 14 onwards...." - data in the corresponding Figure 2A mentions PCW 12.

Answer: We thank the reviewer for this comment. To clarify this, we have adapted this section to: *"From PCW 14 onwards, AMH, like SOX9, became progressively restricted to the basal region, suggesting the onset of Sertoli cell polarisation during this developmental window. This basal restriction became progressively more pronounced with advancing foetal age (Fig 2A (PCW 21, yellow arrowheads) and Fig S1B), with AMH expression remaining consistently high up to the oldest examined foetal age of PCW 21 (Fig 2A, C, Supplementary table S3 and S4)."* in L166-170.

In addition, we have also included a *Supplementary figure S1B* including foetal testis at PCW 14 and 21, showing SOX9 (in magenta) and AMH (in yellow). This figure highlights the onset of AMH co-localisation and its progressive basal restriction during foetal testis development.

c. Rather than reiterating study findings, the Discussion should place the results in context with existing literature on primate and rodent testicular development. The authors should also discuss study limitations, including discuss the limitations including sample number, inter-sample variability, and potential implications for interpretation.

Answer: We acknowledge the significant contributions that primate and rodent studies have made to understanding testicular development, and we have dedicated a paragraph that describes references to this foundational work in the introduction section of our manuscript to provide context for our human-specific findings (L69-80). Due to the significant differences in developmental timelines between rodents and humans, we have chosen to focus our discussion primarily on human studies, including both our own data and findings from other researchers on human testis maturation. Regarding study limitations, we have now added two dedicated paragraphs in the discussion addressing both the limited sample size and the constraints inherent to our methodology (L326-333 and L353-363).

2. Methodology: As stated above, the authors must clearly specify sample numbers, replicates, and p-values for all quantitative data.

* Several claims are made without supporting data. For instance, Lines 121-122: "... with spermatocytes and spermatids present around 10 and 13.5 years old respectively"

lines 163-164:"...AR was not detected in Sertoli cells during early postnatal period but began to increase gradually from around 4 years of age....".

If these statements refer to supplemental data, cite them explicitly in the text.

Answer: We thank the reviewer for these comments. To clarify the first point, we have revised the sentence to - *"The range of developing germ cell types observed increased with age, with spermatocytes and spermatids appearing around 10 and 13.5 years old respectively (Supplementary table S2)."* in L138-141. To support this, we have added a column indicating the most advanced germ cell type observed in each prepubertal sample to Supplementary table S2.

Regarding the second point, we have now included a *Supplementary figure S1C*, which illustrates AR expression in Sertoli cells from 2, 4 and 8-year-old testis sample, with SOX9 (in magenta) and AR (in cyan), which is referred to in L187. In the figure, SOX9+/AR+ double positive Sertoli cells are indicated by white arrowheads, thereby providing direct visual evidence for the gradual increase of AR expression in Sertoli cells from around 4 years of age.

* Lines 136-137: Clarify how did germ cells were visualized. Were all six reported markers applied on the same tissue sections in multi-IF staining?

Answer: Thank you for noticing this. The six markers were divided across two panels (SOX9 x AMH x AR and SOX9 x OCT3/4 x MAGE-A4 x Ki67) and serial sections were used for the respective staining rather than all markers being applied to the same tissue section. This has been clarified in the methods section as *‘Consecutive 5-µm sections were stained for Sertoli cell and germ cell markers respectively to allow direct comparison of Sertoli cell and germ cell maturation and proliferation for each sample’*. in L455-456.

To clarify this further, the sentence is now updated to: *“From approximately PCW 14 onward, a clear reorganisation of the seminiferous cords became apparent as the Sertoli cell nuclei, marked by SOX9 expression, were increasingly aligned along the basal membrane(Fig 2A (PCW 21, magenta arrowheads), Fig S1B).”* in L152-156.

* Line 187: MAGE-A4 expression is shown at PCW 11 in Figure 3A, but text mention PCW 9. In Figure 3C quantification shows variable expression across different fetal PCWs. Please clarify.

Answer: We thank the reviewer for noticing this important discrepancy. We accidentally used an earlier version of the graph during the initial submission that did not reflect submitted final counts for this specific time point as correctly submitted in *Supplementary table S3*. We have now corrected the graph to match the final counts, and the corresponding text to *“Notably, MAGE-A4+ pre-spermatogonia were first observed sporadically within the testicular cords until PCW 10 (Fig 3A, Supplementary table S4 and S5)”* in L211-213, to ensure consistency with the updated quantification data.

* Line 198-199: The text states a "gradual increase in the number of both OCT3/4+ gonocytes and MAGE-A4+ pre-spermatogonia per tubular cross section with advancing gestational age." However, Figure 3C shows the OCT3/4 increasing between PCW 8 to 10 and MAGE-A4 after PCW14. Given this variability, clarify whether the differences are statistically significant.

Answer: We agree with the reviewer that this interpretation does not match the expression of OCT3/4 as seen in Figure 3D. Therefore, we have removed this sentence in the manuscript.

* State the rationale clearly before reporting the experimental observations in the result section.

Answer: We have made alterations to the introduction clarifying our rationale behind our work (L56-58 and 107-110), namely: *“Yet, important gaps in knowledge remain regarding the timing of these early events, which limit progress in developing strategies for fertility preservation and restoration, in children requiring gonadotoxic treatments (Duffin et al., 2024a).”* and *“This analysis, based on the most extensive human testicular sample collection reported to date, provides crucial insights into early testicular development, and lays the foundation for improving fertility preservation for prepubertal male patients at risk of losing their fertility.”* respectively.

Minor edits:

1. Avoid abbreviations in the abstract.

Answer: We have now included the expanded names of the factors - anti-Müllerian hormone (AMH), Androgen Receptor (AR), Octamer binding transcription factor (OCT) 3/4 and Melanoma associated antigen (MAGE)-A4 in L32-37 of the abstract.

2. Lines 42: "Male reproductive capacity....." Specify human/primate.

Answer: To clarify and keep the focus on human testicular development, we have specified that this is in humans in L49.

3. Line 93-97: The paragraph end abruptly-consider summarizing the study outcomes and implications.

Answer: We thank the reviewer for this comment. We have now rephrased the ending to - *“This analysis, based on the most extensive human testicular sample collection reported to date, provides crucial insights into early testicular development, and lays the foundation for improving fertility preservation for prepubertal male patients at risk of losing their fertility.”* in L107-110.

4. Line 52; Expand "SRY"

Answer: We have expanded sex-determining region Y (SRY) in L60.

5. Line 126: Remove "%"

Answer: The % has been removed in L145.

6. Line 169: Replace "LH" with "Luteinizing Hormone"

Answer: We have included the expansion of LH in L347-348.

7. Line 338: Change "n=14, 2-13,5 years old" to "n=14, 2-13.5 years old"

Answer: We have corrected as suggested by the reviewer in L400.

8. Line 396: Clarify whether the 8 images correspond to the same section or Z-plane, and indicate section depth if applicable.

Answer: We have clarified this in L508-509 and added the explanation of consecutive sections used in L455-456.

9. Line 712: Replace "Haematoxylin" with "Hematoxylin"

Answer: We replaced Haematoxylin with Hematoxylin in L433, 439 and 840.

10. Line 712 and 714: Replace "HE" with "H&E"

Answer: We replaced HE with H&E in L840.

11. Line 715: Replace "6,5" with "6.5"

Answer: We have corrected as suggested by the reviewer in L843.

12. Replace "Qupath" with "QuPath"

Answer: We replaced Qupath to QuPath in L511, 549 and 553.

13. Include Y-axis units for Figures 2B and D

Answer: We have included the units for Y axis in the figures, wherever applicable.

Extra changes: In addition to all the changes described above in the answers to the two reviewers, we have also incorporated a number of additional changes to enhance the clarity and readability of the entire manuscript. These changes are also indicated in blue and are underlined.

Second decision letter

MS ID#: bio.062319R1

MS Title: Temporal dynamics of Sertoli and germ cell development in human foetal and prepubertal testis

Authors: Iris Sanou; Mathangi Lakshmipathi; Lisette K. Schönhage; Saskia K. M. van Daalen; Cindy M. de Winter-Korver; Andreas Meißner; Geert Hamer; Dirk G. de Rooij; Rod T. Mitchell; Callista L. Mulder

Article Type: Research Article

Dear Dr Mulder,

I am happy to tell you that your manuscript has been accepted for publication in Biology Open, pending our standard publication integrity checks. It was accepted on 5 January 2026.